# INVERSE GFLOWNETS FOR GENERATIVE IMITATION LEARNING

## ABSTRACT

Sequential generative models are typically trained by maximizing the evidence lower bound (ELBO), which optimizes the likelihood of predicting the next observation given the current one. While ELBO-based training is simple and scalable, in sequential settings it suffers from compounding errors. In this work, we reinterpret ELBO training as an imitation learning problem for modeling data distributions. We show that prior formulations suffer from an entropy bias that is misaligned with the objectives of generative modeling. To address this issue, we leverage the GFlowNet framework to eliminate the bias and derive algorithms that can be viewed as regularized ELBO objectives. Our approach assigns positive rewards to data samples and negative rewards to policy-generated samples, corresponding to minimization of the $\chi^2$-divergence between the data distribution and the policy mixture. We further establish theoretical connections to existing imitation learning methods, providing transferable insights across domains. Empirically, our approach eliminates entropy bias and achieves improved performance on a range of generative modeling tasks by combining with previous methods.

## 1 INTRODUCTION

Framing generative modeling through the lens of imitation learning offers several advantages. For example, recent work has applied maximum entropy inverse reinforcement learning (MaxEnt IRL) to language modeling (Cundy & Ermon, 2023; Wulfmeier et al., 2024), which has been shown theoretically to mitigate compounding errors (Xu et al., 2020). However, unlike in language modeling, where maximum likelihood estimation (MLE) naturally aligns with an imitation learning perspective by treating each data sequence as a unique demonstration, there is no equally principled imitation learning framework for generative models trained by maximizing evidence lower bound (ELBO). In ELBO-based training, the model passes through a series of intermediate objects, with many possible paths leading to the same final outcome. These intermediate objects are not directly observed in the data but are introduced as auxiliary constructs that provide the structure needed to generate objects.

We argue that previous imitation learning based on MaxEnt IRL is not directly applicable to ELBO-based training. First, in MaxEnt IRL the policy is encouraged to select actions uniformly, which can bias the sampling distribution, as we demonstrate in Section 3.1. Second, existing imitation learning frameworks do not consider the variational distribution, as they assume trajectories are fixed demonstrations provided by an optimal expert. In contrast, ELBO-based training samples trajectories from a variational distribution that can be jointly optimized with the generative model, allowing both components to adapt during training.

In this paper, we develop alternative formulations based on Generative Flow Networks (GFlowNets) (Bengio et al., 2021). GFlowNets have been shown to have close connections to variational inference (Malkin et al., 2022b) and to maximum entropy reinforcement learning (MaxEnt RL) (Tiapkin et al., 2024), which provide useful analytical tools for our work. Similar to MaxEnt IRL, our approach first recovers the reward function inherent in the dataset and then trains a policy using the recovered reward. This perspective unifies several prior methods, ranging from energy-based GFlowNets (EB-GFN) (Zhang et al., 2022) to soft Q imitation learning (SQIL) (Reddy et al., 2019). Building on this framework, we propose algorithms that can be interpreted as regularized ELBO objectives, assigning positive rewards to data samples and negative rewards to policy-sampled data. The main contributions of this paper are as follows:

- We extend the imitation learning framework to ELBO-based sequential generative modeling, where only terminal data points are observed. We show that existing frameworks are misaligned with the objectives of generative modeling and introduce biases into the sampling distribution.

- We establish new theoretical connections between MaxEnt IRL, energy-based models (EBMs), and GFlowNets. In particular, we show that EBMs can be reinterpreted as regularized ELBO objectives, which can be optimized through two competing GFlowNet objectives. This perspective further reveals an equivalence to MaxEnt IRL with an additional posterior regularization term.

- We empirically demonstrate the versatility of our framework across diverse generative modeling tasks, and show how previous approaches can be adapted and unified within our formulation.

## 2 BACKGROUND

### 2.1 GENERATIVE FLOW NETWORKS

Given a state space $\mathcal{S}$ and a set of terminal states $\mathcal{X} \subset \mathcal{S}$, sequential generative models aim to generate samples from $\mathcal{X}$ by following a sequence of transitions $(s_0, \ldots, s_T)$, where the intermediate states are denoted by $\tau = (s_0, \ldots, s_{T-1})$ and the terminal state by $x = s_T$. The forward dynamics are governed by a policy $\pi$, which specifies the probability of transitioning to the next state. In addition, we define a backward policy $q$ (interpreted as a variational distribution), which samples trajectories in reverse from terminal states, thereby inducing two Markov chains:

$$\pi(x,\tau) := \pi_0(s_0) \prod_{t=1}^{T} \pi(s_t|s_{t-1}), \qquad q(\tau|x) := \prod_{t=1}^{T} q(s_{t-1}|s_t).$$

where we extend the notation $\pi$ and $q$ to also denote their induced trajectory-level distributions. The probability of a terminal state under $\pi$ is obtained by marginalizing all intermediate states $\pi_{\mathcal{X}}(x) = \sum_{\tau} \pi(x,\tau)$. The goal of GFlowNets is to match a given reward function $r : \mathcal{X} \to \mathbb{R}$ such that $\pi_{\mathcal{X}}(x) = \exp(r(x))/Z$, where $Z$ is a normalizing constant. Since the direct evalutaion of $\pi_{\mathcal{X}}(x)$ is generally infeasible, GFlowNets are trained by jointly optimizing $\pi$ and $q$ using the Trajectory Balance (TB) objective:

$$\text{TB}(x,\tau) := \left(\log Z + \log \pi(x,\tau) - \log q(\tau|x) - r(x)\right)^2$$

While the TB objective is widely used for its improved credit assignment capability (Malkin et al., 2022a), it suffers from high gradient variance (Madan et al., 2023) and may become computationally impractical in long-horizon settings. Alternatively, the Detailed Balance (DB) objective is defined at the transition-level $(s, s')$ as

$$\text{DB}(s,s') := \left(\log F(s) + \log \pi(s'|s) - \log q(s|s') - \log F(s')\right)^2$$

where $F$ denotes the learned state-flow function, interpreted as an unnormalized state distribution. For a terminal state $x$, we set $\log F(x) = r(x)$, while for initial state $s_0$, $\log F(s_0)$ is set to $\log Z + \log \pi_0(s_0)$, where $\pi_0$ is the initial distribution. The main result from GFlowNets theory is that TB and DB induces a policy that samples from the distribution with density $\pi_{\mathcal{X}}(x) \propto \exp(r(x))$ (Malkin et al., 2022a), with a analogous result holding in continuous spaces (Lahlou et al., 2023).

In GFlowNets, rewards are defined only on terminal states $\mathcal{X}$, whereas MaxEnt RL specifies per-step rewards $\bar{r} : \mathcal{S} \times \mathcal{S} \to \mathbb{R}$, and trains a policy to sample trajectories in proportion to their cumulative rewards: $\pi(x,\tau) \propto \exp(\sum_{t=1}^{T} \bar{r}(s_{t-1}, s_t))$. However, if the per-step rewards are augmented with $\log q$, MaxEnt RL also induces policies whose terminal distribution satisfies $\pi_{\mathcal{X}} \propto \exp(r(x))$ (Tiapkin et al., 2024). This correspondence allows us to express the GFlowNet procedure as the following maximum entropy optimization problem:

$$\text{GFN}_q(r) = \arg\max_{\pi} \mathbb{E}_{x,\tau \sim \pi(x,\tau)}[r(x) + \log q(\tau|x)] + H(\pi)$$

where $H(\pi) = \mathbb{E}_\pi[-\log \pi(x, \tau)]$ denotes the trajectory-level entropy. The resulting policy samples trajectories according to $\pi(x, \tau) \propto q(\tau|x) \exp(r(x))$, inducing the marginal $\pi_{\mathcal{X}}(x) \propto \exp(r(x))$. Unlike MaxEnt RL, however, GFlowNets also allow the backward policy $q$ to be trained jointly with $\pi$ so as to approximate the posterior $q(\tau|x) \approx \pi(x, \tau)/\pi_{\mathcal{X}}(x)$.

## 2.2 MAXIMUM ENTROPY INVERSE REINFORCEMENT LEARNING

In MaxEnt IRL, the algorithm has access to a set of demonstrations consisting of state-action pairs assumed to be sampled from an expert. For consistency with sequential generative modeling, we assume a finite-horizon, deterministic, no-discount, acyclic setting. The objective is to learn a per-step reward function $\bar{r}$ such that the expert outperforms all other policies, with a convex regularizer $\psi$, by solving the following optimization problem:

$$\max_{\bar{r}} \min_{\pi} \mathbb{E}_{x,\tau \sim \pi_E(x,\tau)} [\bar{r}(x, \tau)] - \mathbb{E}_{x,\tau \sim \pi(x,\tau)} [\bar{r}(x, \tau)] - H(\pi) - \psi(\bar{r}) \tag{1}$$

where $\pi_E$ is the expert policy and $\bar{r}(x, \tau)$ is the cumulative reward. The policy learned from the reward recovered by Equation 1 is given by $\arg\min_\pi \psi^*(\pi_E - \pi) - H(\pi)$, where $\psi^*(x) = \sup_{y \in \mathbb{R}^{S \times S}} x^T y - \psi(y)$ is the convex conjugate of $\psi$ (Ho & Ermon, 2016, Proposition 3.2).

## 2.3 EVIDENCE LOWER BOUND (ELBO)

For a given data distribution $p_{\text{data}}$, the ELBO objective is defined as

$$\mathbb{E}_{x \sim p_{\text{data}}(x), \tau \sim q(\tau|x)}[\log \pi(x, \tau) - \log q(\tau|x)].$$

In language modeling that generates tokens sequentially from left to right, each sequence $x$ corresponds to a unique trajectory. In this case, $\log q(\tau|x) = 0$, and the trajectory probability reduces to the induced state distribution: $\pi(x, \tau) = \pi_{\mathcal{X}}(x)$. Consequently, the ELBO objective coincides with MLE. While MLE and ELBO objectives are widely used for generative modeling, they are prone to compounding errors (Ross et al., 2011). This issue is tied to the divergence they minimize: for fixed $q$, the ELBO objective corresponds to minimizing the KL divergence, $D_{KL}(p_{\text{data}} \cdot q \| \pi)$. Under KL divergence, even when $p_{\text{data}}(x) \approx 0$ but $\pi(x, \tau)$ is large, the loss remains small. As a result, such objectives permit small errors and tend to exhibit mode-covering behavior. Furthermore, because KL divergence trains the model only on finite data in practice, its behavior is left undetermined on out-of-distribution inputs.

## 3 GENERATIVE IMITATION LEARNING WITH GFLOWNETS

### 3.1 ON THE LIMITATIONS OF MAXENT IRL FOR GENERATIVE MODELING

While it may be tempting to apply the MaxEnt IRL framework directly to sequential generative modeling, it introduces bias in the sampling distribution. Figure 1 illustrates this effect in the Pascal's triangle environment. A policy starts from the topmost hexagons and repeatedly chooses between two actions, `left` or `right`, until it reaches a bottom state. The data distribution $p_{\text{data}}$ is uniform over the bottom states, so ideally the policy should also terminate uniformly (Figure 1b). However, a policy trained with SQIL (Reddy et al., 2019) assigns higher probability to the middle states, which admit a larger number of trajectories (Figure 1a). This bias arises from the entropy bonus, which encourages uniform action probabilities and is reflected in the smoother color transitions at the top of the triangle. This issue does not appear in recent work on applying the MaxEnt IRL framework to language modeling (Wulfmeier et al., 2024), since in those tasks each trajectory corresponds to a unique terminal states.

Another limitation of MaxEnt IRL is the absence of a principled framework for learning the backward policy $q(\tau|x)$. Without a learnable backward policy, the model cannot adapt $q$ to approximate the true posterior over trajectories, which restricts its expressiveness and can introduce additional bias in practice (Figure 1c). In contrast, GFlowNets explicitly incorporate the learning of both $\pi$ and $q$, enabling more accurate modeling of the data distribution (Figure 1d).

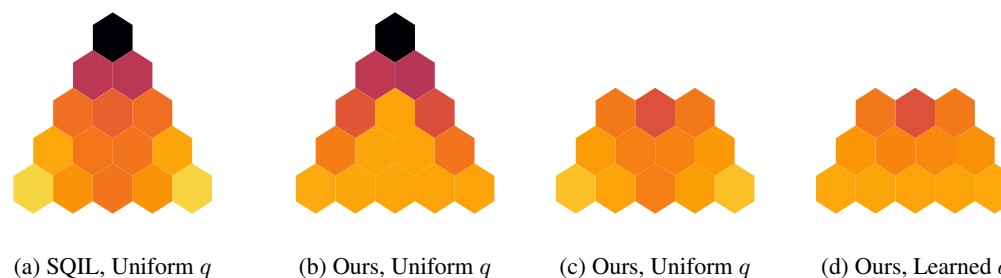

(a) SQIL, Uniform $q$      (b) Ours, Uniform $q$      (c) Ours, Uniform $q$      (d) Ours, Learned $q$

Figure 1: Example experiments in the Pascal's triangle environment, with states colored by visitation probability. **(a, b)** While SQIL concentrates probability mass in the center, our method distributes it uniformly, as desired. **(c, d)** The initial distribution is fixed as $\pi_0(s_0) = [0.3, 0.4, 0.3]$ over the top three states, which limits the modeling capacity of $\pi(x, \tau)$. Nonetheless, jointly learning $\pi$ and $q$ recovers the uniform data distribution at the bottom states.

## 3.2 Generative Imitation Learning Framework

In this section, we derive an generative imitation learning framework based on GFlowNets. Analogous to inverse reinforcement learning, which first recovers the reward function underlying expert demonstrations to train an imitating policy, our formulation relies on reward function that underlies the dataset. Proofs of all Propositions and Lemmas are provided in Appendix A.

Since our focus in this paper is on imitating the data distribution, the rewards assigned to terminal states play a much more critical role than in prior approaches. We therefore derive an alternative formulation in which only the terminal rewards are learned to discriminate between samples from the data and those generated by the policy.

$$\max_r \min_\pi L_q(r, \pi) = \mathbb{E}_{x \sim p_{\text{data}}(x)}[r(x)] - \mathbb{E}_{x, \tau \sim \pi(x, \tau)}[r(x) + \log q(\tau|x)] + H(\pi) - \psi(r)$$

$$= \mathbb{E}_{x \sim p_{\text{data}}(x)}[r(x)] - \mathbb{E}_{x \sim \pi_{\mathcal{X}}(x)}[r(x)] + K(\pi, q) - \psi(r) \quad (2)$$

where $K(\pi, q) = \mathbb{E}_\pi[\log \pi(x, \tau) - \log q(\tau|x)]$. The inner minimization corresponds to the GFlowNet procedure, which induces $\pi_{\mathcal{X}}(x) = \frac{1}{Z} \exp(r(x))$. The overall optimization problem can be interpreted as energy-based model (EBM) training, with $\pi$ as the generative sampler and $-r$ as the energy function. In particular, EB-GFN (Zhang et al., 2022) trains the energy function jointly with a GFlowNet sampler in order to approximate the normalizing constant $Z$. Further details on these connections are provided in Appendix D.

The posterior regularization term $K$ can be decomposed into a posterior KL divergence and the entropy of the terminal distribution: $\mathbb{E}_{\pi_{\mathcal{X}}}[D_{KL}(\pi(\cdot|x) \| q(\cdot|x))] - H(\pi_{\mathcal{X}})$ (See Appendix A.1). In other words, $K(\pi, q)$ encourages the policy to align its posterior with the reference backward distribution $q$, while the entropy term promotes diversity over the outcome space $\mathcal{X}$. To characterize Equation 2, we first establish the convexity of $K$, followed by a proposition describing the policy induced by the recovered reward.

**Lemma 1** (Convexity of $K$) $K(\pi, q)$ *is convex in both $\pi$ and $q$.*

**Proposition 1** (Induced policy under $L_q$) *If $r^\star \in \arg\max_r \min_\pi L_q(r, \pi)$ is the recovered reward, then the policy induced by $r^\star$ is:* $\text{GFN}_q(r^\star) = \arg\min_\pi \psi^*(p_{\text{data}} - \pi_{\mathcal{X}}) + K(\pi, q)$

Proposition 1 shows that the optimization problem seeks a policy whose distribution over terminal states $\mathcal{X}$ closely matches the data distribution, as measured by $\psi^*$, while simultaneously aligning the trajectory distribution with $q$. By choosing an appropriate $\psi$, one can recover well-known statistical divergence measures, as indicated in Appendix B. This formulation cleanly separates the characteristics of the induced policy for terminal states from those for trajectories.

A solution to the max-min optimization can be obtained by iteratively (1) training the GFlowNet policy in the inner loop and (2) learning the rewards in the outer loop, as is done in EB-GFN. However, this approach is challenging in practice due to the adversarial nature of the optimization.

To simplify the optimization, we reparameterize the problem in terms of the policy. Given a reward function, the induced policy is uniquely defined by $\text{GFN}_q(r)$. Conversely, the reward function can be expressed in terms of the policy $\pi$ and the normalization constant $Z$:

$$r_\pi(x) = \log Z + \log \mathbb{E}_{\tau \sim q(\tau|x)}[\pi(x, \tau)/q(\tau|x)].$$

where $r_\pi$ is the reward function reparameterized by the policy $\pi$. This allows the reward to be represented through the pair $(\pi, Z)$, leading to the reformulated objective: $\max_{\pi, Z} \min_{\tilde{\pi}} L_q(r_\pi, \tilde{\pi})$. Using this new representation, the solution to the inner minimization is given by $\tilde{\pi}(x, \tau) = \pi_\mathcal{X}(x)q(\tau|x)$. This removes the need for a separate inner-loop optimization, as we can directly sample $x \sim \pi_\mathcal{X}(x)$ and $\tau \sim q(\tau|x)$.

However, the inner expectation required to compute $r_\pi(x)$ remains difficult to evaluate accurately. To address this, we approximate $r_\pi(x)$ by its single-sample estimate $\hat{r}_\pi(x; \tau) = \log Z + \log \pi(x, \tau) - \log q(\tau|x)$, which serves as a trajectory-level estimate of the implicit reward $r_\pi(x)$ under policy $\pi$. This leads to the following approximate objective:

$$\max_{\pi, Z} \min_{\tilde{\pi}} \hat{L}_q(\pi, Z, \tilde{\pi}) = \mathbb{E}_{\substack{x \sim p_{\text{data}}(x) \\ \tau \sim q(\tau|x)}} [\hat{r}_\pi(x; \tau)] - \mathbb{E}_{x, \tau \sim \tilde{\pi}(x, \tau)} [\hat{r}_\pi(x; \tau)] + K(\tilde{\pi}, q) - \psi(\hat{r}_\pi) \quad (3)$$

The inner minimization is attained at $\tilde{\pi}(x, \tau) = \pi(x, \tau)$ (Lemma 2 in Appendix), eliminating the need for the inner optimization loop. The resulting objective only depends on $\pi$ and $Z$ as follows:

**Proposition 2** (Regularized ELBO) *Denoting $\mathcal{J}_q(\pi, Z) = \min_{\tilde{\pi}} \hat{L}_q(\pi, Z, \tilde{\pi})$, we obtain*

$$\mathcal{J}_q(\pi, Z) = \underbrace{\mathbb{E}_{x \sim p_{\text{data}}(x), \tau \sim q(\tau|x)} [\log \pi(x, \tau) - \log q(\tau|x)]}_{\text{ELBO}} \underbrace{- \psi(\hat{r}_\pi)}_{\text{regularization}} \quad (4)$$

*recovering the ELBO objective with an additional regularization term.*

The regularization term constrains the reward, which is implicitly defined by the policy. From Jensen's inequality, $r_\pi(x) \geq \mathbb{E}_{\tau \sim q(\tau|x)}[\hat{r}_\pi(x; \tau)]$, and that the single sample approximation $\hat{r}_\pi$ becomes accurate when $q(\tau|x) \propto \pi(x, \tau)$. Therefore, the approximation error can be reduced by interleaving optimization steps for $q$ with those for $p$. However, optimizing $\mathcal{J}_q(\pi, Z)$ with fixed $q$ also results in $\pi(x, \cdot) \propto q(\cdot|x)$, as shown in the following proposition.

**Proposition 3** (Induced policy under $\mathcal{J}_q$) *Maximizing $\mathcal{J}_q$ is equivalent to minimizing a divergence regularized by $K(\pi, q)$: $\arg\max_\pi \max_Z \mathcal{J}_q(\pi, Z) = \arg\min_\pi \psi^*(p_{\text{data}} \cdot q - \pi) + K(\pi, q)$.*

For proper choices of $\psi$, the conjugate $\psi^*$ induces a divergence that encourages $\pi(x, \cdot) \propto q(\cdot \mid x)$ (Appendix B). Since $K$ promotes the same alignment, the approximate objective remains accurate provided that $\pi$ is optimized over a sufficiently large function class. Because both $\psi^*$ and $K$ are convex, the resulting objective ensures stable optimization, leading to the unique optimal point.

**Remark** Our derivation is closely related to IQ-Learn (Garg et al., 2021), which also eliminates the inner optimization loop by reparameterizing both the reward function and the policy in terms of the soft-$Q$ function. IQ-Learn can be interpreted as a form of regularized MLE (Wulfmeier et al., 2024), which parallels the regularized ELBO objective presented in Proposition 2. In MaxEnt IRL, however, the policy is encouraged to maximize $H(\pi)$, which biases it toward terminal states with many trajectories leading to them. In contrast, our objectives eliminate this bias by leveraging the GFlowNets perspective, which was itself motivated by the same issue.

### 3.3 DERIVING GFLOWNETS OBJECTIVES

While various convex functions can be chosen for the regularizer $\psi$, we adopt $\psi_{\text{TB}}(\hat{r}_\pi) = \alpha \mathbb{E}_{d_{\text{mix}}}[(\hat{r}_\pi(x; \tau) - r_{\text{prior}}(x))^2]$, where $r_{\text{prior}}(x)$ is the prior rewards we have on $\mathcal{X}$, $d_{\text{mix}} = \frac{1}{2}(p_{\text{data}} \cdot q + \pi)$ is the mixture distribution between $p_{\text{data}} \cdot q$ and $\pi$, and $\alpha$ controls the strength of the regularization. Under this choice, Equation 4 reduces to a mixed objective (ELBO + TB). In fact, it can be reformulated entirely as two competing TB objectives, which we term TBIL.

**Proposition 4** (TBIL) *The solution to Equation 4 with the regularizer $\psi_{\mathrm{TB}}$ is equivalent to solving two TB objectives. Specifically, we define $\mathcal{L}_q^{\mathrm{TB}}(\pi, Z)$ as:*

$$\underset{\substack{x \sim p_{\mathrm{data}}(x) \\ \tau \sim q(\tau|x)}}{\mathbb{E}} \left[ (\hat{r}_\pi(x; \tau) - r_{\mathrm{prior}}(x) - r_\alpha)^2 \right] + \underset{x, \tau \sim \tilde{\pi}(x, \tau)}{\mathbb{E}} \left[ (\hat{r}_\pi(x; \tau) - r_{\mathrm{prior}}(x) + r_\alpha)^2 \right], \quad (5)$$

*where $r_\alpha = 1/\alpha$, and $\tilde{\pi}$ denotes the sampling distribution. If the samples are drawn from the current policy, i.e., $\tilde{\pi} = \pi$, then $\arg\min_{\pi, Z} \mathcal{L}_q^{\mathrm{TB}}(\pi, Z) = \arg\max_{\pi, Z} \mathcal{J}_q(\pi, Z)$.*

The resulting objective assigns different rewards to samples depending on their source: samples from the data distribution receive rewards of $r_{\mathrm{prior}}(x) + r_\alpha$, while other samples receive $r_{\mathrm{prior}}(x) - r_\alpha$. When $r_\alpha = 0$, the objective reduces to the TB objective. For $r_\alpha > 0$, the objective progressively incorporates the ELBO term. The equivalence between the regularized ELBO and Equation 5 can be established more directly through a gradient analysis, which we provide in Appendix C.

The prior reward function $r_{\mathrm{prior}}$ can be leveraged to incorporate domain knowledge while imitating the data distribution. For example, Pandey et al. (2025) combined the negative ELBO loss with the TB loss to pretrain a molecular generative model. In this setting, the negative ELBO encourages proximity to the reference chemical library, while the prior is defined through computationally inexpensive reward functions such as drug-likeness (QED), which act as proxies for more complex molecular properties. Our results show that this regularized ELBO objective is equivalent to two competing TB objectives for specific choices of $\alpha$ and the mixture proportion in $d_{\mathrm{mix}}$.

While Equation 5 involves two different TB objectives with different rewards, it can be shown to be equivalent to minimizing the $\chi^2$-divergence, which attains a unique minimum and makes out-of-distribution samples more costly than KL divergence.

**Proposition 5** (Divergence interpretation) *When $r_{\mathrm{prior}}$ is a constant function and $\alpha = 1/2$, minimizing $\mathcal{L}_q(\pi, Z)$ is equivalent to minimizing the $\chi^2$-divergence with additional $K$ term. Specifically, $\arg\min_\pi \min_Z \mathcal{L}_q(\pi, Z) = \arg\min_\pi 2\chi^2(p_{\mathrm{data}} \cdot q \| \frac{1}{2}(p_{\mathrm{data}} \cdot q + \pi)) + K(\pi, q)$.*

Next, we derive analogous results for the DB objective, which is defined over transitions. Just as Equation 5 can be interpreted as the ELBO combined with the TB objective, combining the ELBO with the DB objective yields a pair of competing DB objectives, refered to as DBIL:

**Proposition 6** (DBIL) *Let $\rho_q(s, s')$ and $\rho_\pi(s, s')$ denote distibutions over transitions, induced by $p_{\mathrm{data}} \cdot q$ and $\pi$, respectively. Define the transition-based objective $\mathcal{L}_q^{\mathrm{DB}}$ as*

$$\underset{s, s' \sim \rho_q(s, s')}{\mathbb{E}} \left[ \left( \log \frac{F(s)\pi(s'|s)}{F(s')q(s|s')} - r_\alpha \right)^2 \right] + \underset{s, s' \sim \rho_\pi(s, s')}{\mathbb{E}} \left[ \left( \log \frac{F(s)\pi(s'|s)}{F(s')q(s|s')} + r_\alpha \right)^2 \right], \quad (6)$$

*where $r_\alpha$ is some constant, and $\log F(x) = r_{\mathrm{prior}}(x)$ for terminal states $x$. The objective $\mathcal{L}_q^{\mathrm{DB}}$ is equivalent to the negative ELBO combined with the DB objective. Moreover, when $r_{\mathrm{prior}}(x)$ is constant and $r_\alpha = 2$, optimizing $\mathcal{L}_q^{\mathrm{DB}}$ is equivalent to minimizing $\chi^2(\rho_q \| \frac{1}{2}(\rho_q + \rho_\pi)) + k(\pi, q)$, where $k(\pi, q) = \mathbb{E}_{\rho_\pi}[\log \pi(s'|s) - \log q(s|s')]$ is a per-step regularization term.*

DBIL assigns a $(\pm r_\alpha)$ bonus at each transition, which can be understood as an energy cost (or gain) associated with the transition (Pan et al., 2023; Jang et al., 2023).

So far, we have assumed the backward policy $q$ to be fixed, aligning only $\pi$ with $q$. While this suffices to recover the correct distributions if the policy class is expressive enough, jointly learning $q$ can yield faster convergence (Malkin et al., 2022a) and improved performance in several domains using ELBO-based objectives (Chen et al., 2021; Sahoo et al., 2024). An important advantage of TBIL and DBIL, paralleling GFlowNet objectives, is that $\pi$ and $q$ can be trained jointly, thereby eliminating the need for separate optimization steps.

**Remark** The resulting algorithm is similar to SQIL (Reddy et al., 2019), which uses fixed zero-one rewards to the policy and the expert at each transition. SQIL has also been shown to connect to the $\chi^2$-divergence when symmetric rewards are used (Al-Hafez et al., 2023). In fact, through the established connections between GFlowNets and MaxEnt RL (Tiapkin et al., 2024), $\log F + \log \pi$ can be interpreted as a soft-$Q$ function, making the two algorithms closely related. The key distinction, however, is that Equation 6 explicitly incorporates $\log q$ as a reward baseline.

### 3.4 PRACTICAL ALGORITHM

To instantiate our algorithm, we approximate $\pi$, $Z$ (for TBIL), $F$ (for DBIL), and optionally $q$ with neural networks, while expectations are estimated using finite samples. An overview of TBIL is presented in Algorithm 1. Although the proposed algorithms require samples from the policy, off-policy training can be performed with a replay buffer, which has been shown to substantially improve performance (Du & Mordatch, 2019; Kostrikov et al., 2018).

---

**Algorithm 1** TBIL

**Require:** Dataset $\mathcal{D}$, $r_\alpha$, $\pi_\theta$, $q_\phi$, and $Z_\gamma$
1: Initialize parameters $\theta, \gamma$ and optionally $\phi$
2: **while** not converged **do**
3:     Sample $(x, \tau)$ from $\mathcal{D}$ with $\tau \sim q_\phi(\tau | x)$
4:     Sample $(x', \tau') \sim \pi_\theta(x, \tau)$
5:     Update $\theta, \phi, \gamma$ using Equation 5
6: **end while**

---

While we can assume a fixed horizon length $T$ without loss of generality by introducing an absorbing state, in practice no further interactions are needed once termination is reached. For DBIL, however, this setup can cause longer trajectories to accumulate larger cumulative $r_\alpha$ bonuses, thereby introducing bias. To correct for this, we assign an additional reward by setting $\log F(x)$ as $r_{\text{prior}}(x) \pm (T - t)r_\alpha$ when a trajectory ends at step $t$. Although this issue has been noted previously in the imitation learning literature (Kostrikov et al., 2018), we revisit it here in the context of generative modeling and provide further discussion in Appendix E.

## 4 RELATED WORK

**Imitation Learning** Early approaches such as behavioral cloning (BC) treat imitation learning (IL) as supervised learning over expert state-action pairs, but they suffer from compounding errors due to distributional shift (Ross & Bagnell, 2010; Ross et al., 2011). To mitigate this issue, inverse reinforcement learning (IRL) methods jointly infer both the policy and the reward function, which has been shown to reduce compounding errors (Xu et al., 2020). In particular, GAIL (Ho & Ermon, 2016) formulates IL as adversarial training between the policy and the reward function, and shows that minimizing the divergence between expert and policy occupancy measures can be expressed as a two-player saddle-point problem. IQ-Learn (Garg et al., 2021) represents both the policy and rewards using a soft $Q$-function, eliminating the need for adversarial training. SQIL (Reddy et al., 2019) simplifies IL by showing that a zero-one reward scheme is equivalent to a form of regularized BC, which was later connected to the general IL framework (Al-Hafez et al., 2023). Connections between IRL and EBMs were established in Finn et al. (2016).

**GFlowNets** GFlowNets were introduced as a framework for training policies that sample compositional objects in proportion to a given reward function (Bengio et al., 2021). Subsequent work has highlighted their close connections to variational inference (Malkin et al., 2022b; Zimmermann et al., 2023) and MaxEnt RL (Tiapkin et al., 2024; Mohammadpour et al., 2024), which provide useful theoretical tools for analysis. Since their introduction, GFlowNets have been extended to continuous spaces (Lahlou et al., 2023) and to environments beyond directed acyclic graphs (Brunswic et al., 2024; Morozov et al., 2025), developments that are complementary to and potentially extend our work. Most similar to our work, Zhang et al. (2022) proposed training a GFlowNet sampler to aid energy model learning on a given dataset, a procedure that can be viewed as interleaving the max-min optimization steps in our framework. We provide a more detailed discussion of Zhang et al. (2022) in Appendix D.

## 5 EXPERIMENTS

In this section, we compare three methods that are closely related to our approach—SQIL, EBMs, and GFlowNets. Our aim is not to identify the best-performing method, but to demonstrate how these existing approaches can be adapted within our framework and to provide meaningful comparisons with our proposed method. Further experimental details and results are presented in Appendix F.

### 5.1 GENERATIVE IMITATION LEARNING (SQIL)

Table 1: Negative log-likelihood (NLL ↓) on seven 2D synthetic problems.

| Method | 2spirals | 8gaussians | circles | moons | pinwheel | swissroll | checkerboard |
|--------|----------|------------|---------|-------|----------|-----------|--------------|
| EB-GFN | **20.098** | 20.025 | 20.576 | 19.764 | 19.629 | 20.185 | 20.716 |
| TBIL | 20.131 | **19.998** | 20.586 | 19.774 | 19.639 | 20.194 | **20.712** |
| Combined | 20.106 | 20.002 | **20.575** | **19.759** | **19.612** | **20.180** | 20.721 |

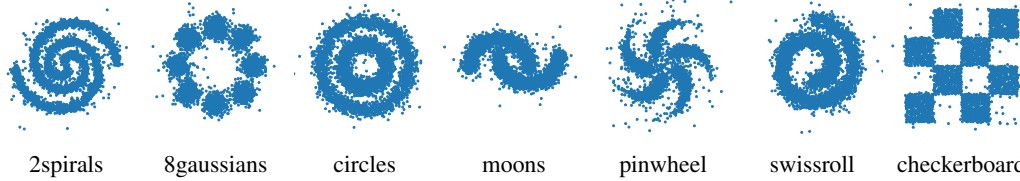

2spirals     8gaussians     circles     moons     pinwheel     swissroll     checkerboard

Figure 3: Samples generated by a GFlowNet trained under the TBIL objective with $r_\alpha = 10$.

We compare DBIL with SQIL (Reddy et al., 2019), which can be viewed as DBIL without the $\log q$ baseline. For fairness, we replaced SQIL's zero–one rewards with symmetric rewards $r_\alpha = \pm 1$, and we also implemented a variant of SQIL augmented with the $\log q$ baseline. The task is to generate a 17-bit binary sequence by flipping one bit at a time until the `stop` action is chosen, with the final bit reserved for this termination signal ($|\mathcal{X}| = 65{,}536$). Following Malkin et al. (2022a), data samples are constructed by concatenating four randomly chosen blocks from the set `0100, 1100, 0110, 0011, 1110`, yielding 625 data samples. Figure 2 reports the convergence of each method, measured by the probability of terminating at states contained in $p_{\text{data}}$. Both DBIL and the corrected SQIL variant successfully learn to match the data distribution, while SQIL ($r_\alpha = 1$) degenerates to always pro-

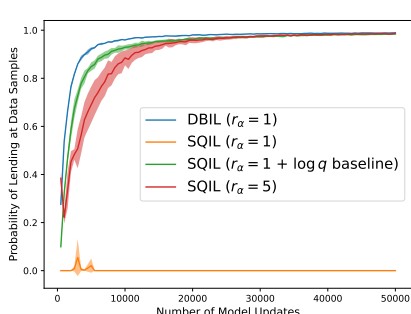

Figure 2: Comparison to SQIL as measured by the probability of sampling data samples.

ducing the all-ones sequence, which is favored due to the factorially larger number of trajectories leading to it. In addition, DBIL and the combined variant generate blocks with approximately uniform frequencies, closely matching the target distribution, whereas SQIL with $r_\alpha = 5$ favors blocks containing many `1`'s.

## 5.2 ENERGY-BASED MODELING (EB-GFN)

We compare TBIL with EB-GFN (Zhang et al., 2022), as both solve the same optimization problem but through different algorithms (see Appendix D for details). Following the experimental setup of Zhang et al. (2022), we use seven target distributions over 32-dimensional binary vectors derived from discretizing continuous distributions on the 2D plane. Each point $(x, y) \in \mathbb{R}^2$ is quantized into $2^{16}$ equal-width bins per coordinate and encoded using a 16-bit Gray code, ensuring that adjacent bins differ by exactly one bit. Default hyperparameters from EB-GFN are used for both methods, except that EB-GFN is trained with an L2 regularization coefficient of $\alpha = 0.1$, corresponding to $r_\alpha = 10$ in TBIL. Since our method can be readily integrated with EB-GFN, we additionally tested a combined approach, using intermediate hyperparameters of $\alpha = 0.2$ and $r_\alpha = 5$. Unlike EB-GFN, TBIL does not require a separate reward network, resulting in fewer effective parameters. Figure 3 presents samples generated by TBIL.

We evaluate each methods in terms of negative log-likelihood (NLL) in Table 1. EB-GFN achieved slightly better overall performance than TBIL at convergence, which may be explained by the greater flexibility offered by explicitly modeling the reward function, a benefit that has also been observed in language modeling tasks (Xu et al., 2024; Ivison et al., 2024). However, because EB-GFN relies solely on the reward function to guide GFlowNet training—which is particularly unreliable in the early stages—it converges more slowly (see Figure 5 in the Appendix). Incorporating TBIL into

Table 2: Results on two biological sequence generation tasks: DNA (TFBind10) and molecules (sEH). We report true reward means for different top-$k$ samples selected from 5,000 model-generated samples (mean $\pm$ std).

| Task | Dataset | $r_\alpha$ | $k = 50$ | $k = 500$ | $k = 5000$ | Diversity |
|---|---|---|---|---|---|---|
| TFBind10 | Top 5% | 0 | $1.047 \pm 0.078$ | $0.603 \pm 0.018$ | $0.045 \pm 0.008$ | $\mathbf{6.497 \pm 0.003}$ |
| | | 0.5 | $\mathbf{1.087 \pm 0.036}$ | $\mathbf{0.621 \pm 0.003}$ | $\mathbf{0.057 \pm 0.002}$ | $6.384 \pm 0.013$ |
| | Top 15% | 0 | $0.906 \pm 0.019$ | $0.597 \pm 0.003$ | $0.071 \pm 0.026$ | $6.403 \pm 0.014$ |
| | | 0.5 | $\mathbf{0.931 \pm 0.020}$ | $\mathbf{0.600 \pm 0.004}$ | $\mathbf{0.093 \pm 0.006}$ | $\mathbf{6.412 \pm 0.008}$ |
| sEH | Top 5% | 0 | $7.906 \pm 0.028$ | $7.465 \pm 0.034$ | $5.425 \pm 0.066$ | $\mathbf{0.783 \pm 0.001}$ |
| | | 0.5 | $\mathbf{7.912 \pm 0.013}$ | $\mathbf{7.487 \pm 0.001}$ | $\mathbf{5.643 \pm 0.036}$ | $0.779 \pm 0.002$ |
| | Top 15% | 0 | $\mathbf{7.840 \pm 0.022}$ | $\mathbf{7.442 \pm 0.009}$ | $5.609 \pm 0.032$ | $0.772 \pm 0.001$ |
| | | 0.5 | $7.818 \pm 0.016$ | $7.412 \pm 0.010$ | $\mathbf{5.700 \pm 0.009}$ | $\mathbf{0.776 \pm 0.003}$ |

the training, as in the combined method, substantially accelerates convergence and achieves the best overall performance.

### 5.3 OFFLINE LEARNING (GFLOWNET)

Existing GFlowNets are typically trained with an oracle function assumed to provide reliable rewards. In practice, however, this oracle is often replaced with a learned proxy model, which may not faithfully capture the true reward (Zhang et al., 2025). This limitation is especially pronounced in domains such as biological sequence generation, where experimental data is scarce and proxy models must be trained on limited datasets, increasing the risk of inaccuracies. A natural remedy is to leverage previously collected data to constrain the policy distribution, thereby improving robustness (Nair et al., 2020). We adopt this approach in our experiments by training conservative GFlowNets that stay close to the offline data distribution. In our formulation, this corresponds to setting $r_\alpha > 0$, which acts as a conservatism parameter.

We evaluate this idea on two generative tasks: DNA (TFBind10) and molecules (sEH). Both tasks can be formulated as sequence construction problems under a prepend–append action space. To assess the impact of offline data quality, we construct datasets by randomly sampling 1000 objects from the top 5% and 15%, which are then used both to train proxy models and as data samples. Table 2 reports the mean rewards of the top-scoring samples generated by different methods. Setting $r_\alpha = 0$ corresponds to standard GFlowNets trained solely on the proxy model, while $r_\alpha = 0.5$ corresponds to our conservative variant. The results indicate that our method improves upon the proxy-only baseline overall, though its effectiveness depends on the quality of the dataset. Further results and detailed definitions of the evaluation metrics are provided in Appendix F.3.

## 6 CONCLUSION

We introduced a generative imitation learning framework built on GFlowNets, extending MaxEnt IRL to settings where a variational distribution is introduced. Our analysis established theoretical links between regularized ELBO and GFlowNet objectives, showing that the regularized ELBO can be reformulated as two competing GFlowNet objectives. The framework naturally supports joint training of forward and backward policies and avoids the entropy bias inherent in prior approaches. We demonstrated that the proposed objectives can be seamlessly integrated into existing methods, broadening their applicability to a variety of generative modeling settings. We conducted experiments on both synthetic and biological sequence design tasks, demonstrating promising results and showing that our approach can be effectively combined with existing methods. However, its empirical validation remains limited. Extending TBIL and DBIL to more complex domains such as molecular graphs or high-dimensional images would better test scalability and practical utility, and represents a key direction for future work.

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

# A PROOFS

## A.1 LEMMA 1

Here we establish the convexity of $K(\pi, q) = \mathbb{E}_\pi[\log \pi(x, \tau) - \log q(\tau|x)]$. Define the posterior distribution $\pi(\tau|x) = \pi(x, \tau)/\pi_\mathcal{X}(x)$. Then $K$ can be decomposed as follows:

$$
\begin{aligned}
K(\pi, q) &= \mathbb{E}_{x,\tau \sim \pi(x,\tau)}[\log \pi(x, \tau) - \log q(\tau|x)] \\
&= \mathbb{E}_{x,\tau \sim \pi(x,\tau)}[\log \pi(\tau|x) + \log \pi(x) - \log q(\tau|x)] \\
&= \mathbb{E}_{x \sim \pi_\mathcal{X}(x), \tau \sim \pi(\tau|x)}[\log \pi(\tau|x) - \log q(\tau|x)] + \mathbb{E}_{x \sim \pi_\mathcal{X}(x)}[\log \pi(x)] \\
&= \mathbb{E}_{x \sim \pi_\mathcal{X}(x)}[D_{KL}(\pi(\cdot|x)||q(\cdot|x))] - H(\pi_\mathcal{X}).
\end{aligned}
$$

where $D_{KL}$ is the KL divergence and $H$ is the entropy. Now let $\pi_\lambda(x, \tau) = \lambda \pi_1(x, \tau) + (1 - \lambda)\pi_2(x, \tau)$ for some $\pi_1$, $\pi_2$ and $0 \leq \lambda \leq 1$. Marginalizing out $\tau$, the induced terminal state distribution is $\pi_{\lambda,\mathcal{X}}(x) = \pi_{1,\mathcal{X}}(x) + (1 - \lambda)\pi_{2,\mathcal{X}}(x)$, which establishes the linearity of the expectation with respect to the terminal distribution $\pi_\mathcal{X}$. Since $D_{KL}$ is jointly convex in its arguments and $-H$ is convex, and the expectation is linear, it follows that $K(\pi, q)$ is convex.

## A.2 PROPOSITION 1

From the Equation 2, we have

$$
\begin{aligned}
L_q(\pi, r) &= \mathbb{E}_{x \sim p_{\text{data}}(x)}[r(x)] - \mathbb{E}_{x \sim \pi_\mathcal{X}(x)}[r(x)] - \psi(r) + K(\pi, q) \\
&= \sum_{x \in \mathcal{X}} r(x)(p_{\text{data}}(x) - \pi_\mathcal{X}(x)) - \psi(r) + K(\pi, q)
\end{aligned}
$$

for some fixed backward policy $q$. Since $K$ and $\psi$ are both convex, we have that $L_q(\cdot, r)$ is convex in $\pi$ for all $r$ and $L_q(p, \cdot)$ is concave in all $r$. Therefore, we can exchange $\min$ and $\max$ as in the following:

$$
\max_r \min_\pi L_q(\pi, r) = \min_\pi \max_r L_q(\pi, r) = \min_\pi \psi^*(p_{\text{data}} - \pi_\mathcal{X}) + K(\pi, q).
$$

Let $r^\star \in \arg\max_r \min_\pi L_q(\pi, r)$ and $\pi^\star \in \arg\min_\pi \max_r L_q(\pi, r)$. Then, $(\pi^\star, r^\star)$ is a saddle point of $L_q$, meaning $\pi^\star \in \arg\min_\pi L_q(\pi, r^\star)$.

### A.3 LEMMA 2

**Lemma 2** $\pi = \arg\min_{\tilde{\pi}} \hat{L}_q(\pi, Z, \tilde{\pi})$.

Rewriting $\hat{L}_q(\pi, Z, \tilde{\pi})$, after removing terms that do not depend on $\tilde{\pi}$,

$$
\begin{aligned}
\arg\max_{\tilde{\pi}} -\hat{L}_q(\pi, Z, \tilde{\pi}) &= \arg\max_{\tilde{\pi}} \mathbb{E}_{x,\tau \sim \tilde{\pi}(x,\tau)} \left[ \hat{r}_\pi(x; \tau) \right] - K(\tilde{\pi}, q) \\
&= \arg\max_{\tilde{\pi}} \mathbb{E}_{x,\tau \sim \tilde{\pi}(x,\tau)} \left[ \log \frac{Z\pi(x,\tau)}{q(\tau|x)} \right] - \mathbb{E}_{x,\tau \sim \tilde{\pi}(x,\tau)} \left[ \log \frac{\tilde{\pi}(x,\tau)}{q(\tau|x)} \right] \\
&= \arg\max_{\tilde{\pi}} \mathbb{E}_{x,\tau \sim \tilde{\pi}(x,\tau)} \left[ \log \pi(x,\tau) \right] + H(\tilde{\pi})
\end{aligned}
$$

This is an entropy-regularized maximization problem, a form with a well-known closed-form solution (see, e.g., Haarnoja et al. (2017); Schulman et al. (2017)):

$$
\tilde{\pi}^\star(x,\tau) \propto \exp\left( \log \pi(x,\tau) \right) = \pi(x,\tau)
$$

### A.4 PROPOSITION 2

We want to prove $\mathcal{J}_q(\pi, Z) = \min_{\tilde{\pi}} \hat{L}_q(\pi, Z, \tilde{\pi})$. Rewriting $\hat{L}_q(\pi, Z, \tilde{\pi})$:

$$
\hat{L}_q(\pi, Z, \tilde{\pi}) = \mathbb{E}_{\substack{x \sim p_{\text{data}}(x) \\ \tau \sim q(\tau|x)}} \left[ \log \frac{Z\pi(x,\tau)}{q(\tau|x)} \right] - \mathbb{E}_{x,\tau \sim \tilde{\pi}(x,\tau)} \left[ \log \frac{Z\pi(x,\tau)}{q(\tau|x)} - \log \frac{\tilde{\pi}(x,\tau)}{q(\tau|x)} \right] - \psi(\hat{r}_\pi)
$$

By Lemma 2, minimizing $\hat{L}_q(\pi, Z, \tilde{\pi})$ with respect to $\tilde{\pi}$ yields $\tilde{\pi} = \pi$. Substituting $\tilde{\pi}$ with $\pi$, we obtain:

$$
\begin{aligned}
\hat{L}_q(\pi, Z, \pi) &= \mathbb{E}_{\substack{x \sim p_{\text{data}}(x) \\ \tau \sim q(\tau|x)}} \left[ \log \frac{Z\pi(x,\tau)}{q(\tau|x)} \right] - \mathbb{E}_{x,\tau \sim \pi(x,\tau)} \left[ \log \frac{Z\pi(x,\tau)}{q(\tau|x)} - \log \frac{\pi(x,\tau)}{q(\tau|x)} \right] - \psi(\hat{r}_\pi) \\
&= \mathbb{E}_{\substack{x \sim p_{\text{data}}(x) \\ \tau \sim q(\tau|x)}} \left[ \log \pi(x,\tau) - \log q(\tau|x) \right] - \psi(\hat{r}_\pi) \\
&= \mathcal{J}_q(\pi, Z)
\end{aligned}
$$

as desired.

### A.5 PROPOSITION 3

For convenience, we rewrite Equation 3 below:

$$
\max_{\pi, Z} \min_{\tilde{\pi}} \hat{L}_q(\pi, Z, \tilde{\pi}) = \mathbb{E}_{\substack{x \sim p_{\text{data}}(x) \\ \tau \sim q(\tau|x)}} \left[ \hat{r}_\pi(x; \tau) \right] - \mathbb{E}_{x,\tau \sim \tilde{\pi}(x,\tau)} \left[ \hat{r}_\pi(x; \tau) \right] + K(\tilde{\pi}, q) - \psi(\hat{r}_\pi)
$$

Since $\hat{r}_\pi(x; \tau) = \log Z + \log \pi(x,\tau) - \log q(\tau|x)$ is not restricted in its range as $(\pi, Z)$ varies, the maximization over $(\pi, Z)$ can equivalently be expressed as a maximization over any function $g : \mathcal{T} \to \mathbb{R}$, where $\mathcal{T}$ denotes the trajectory space $(s_0, \ldots, s_T)$, as follows:

$$
= \max_{g} \min_{\pi} \mathbb{E}_{\substack{x \sim p_{\text{data}}(x) \\ \tau \sim q(\tau|x)}} \left[ g(x,\tau) \right] - \mathbb{E}_{x,\tau \sim \pi(x,\tau)} \left[ g(x,\tau) \right] + K(\pi, q) - \psi(g).
$$

Since $K$ is convex and $-\psi$ is concave, we can exchange the order of max-min and derive the divergence form as follows:

$$= \min_{\pi} \max_{g} \mathbb{E}_{\substack{x \sim p_{\text{data}}(x) \\ \tau \sim q(\tau|x)}} [g(x, \tau)] - \mathbb{E}_{x, \tau \sim \pi(x, \tau)} [g(x, \tau)] + K(\pi, q) - \psi(g)$$

$$= \min_{\pi} \max_{g} \sum_{x, \tau} (p_{\text{data}}(x)q(\tau|x) - \pi(x, \tau))g(x, \tau) - \psi(g) + K(\pi, q)$$

$$= \min_{\pi} \psi^*(p_{\text{data}} \cdot q - \pi) + K(\pi, q).$$

Let $\pi^\star, Z^\star = \arg\max_{\pi, Z} \mathcal{J}_q(\pi, Z)$ and $g^\star(x, \tau) = \log Z^\star + \log \pi^\star(x, \tau) - \log q(\tau|x)$. By the saddle point proterty, $g^\star$ is the maximizer of the inner optimization problem:

$$\pi^\star = \arg\min_{\pi} \psi^*(p_{\text{data}} \cdot q - \pi) + K(\pi, q)$$

$$= \arg\min_{\pi} \mathbb{E}_{\substack{x \sim p_{\text{data}}(x) \\ \tau \sim q(\tau|x)}} [g^\star(x, \tau)] - \mathbb{E}_{x, \tau \sim \pi(x, \tau)} [g^\star(x, \tau)] + K(\pi, q) - \psi(g^\star)$$

$$= \arg\max_{\pi} \mathbb{E}_{x, \tau \sim \pi(x, \tau)} [g^\star(x, \tau) + \log q(\tau|x)] - H(\pi)$$

$$= \arg\max_{\pi} \mathbb{E}_{x, \tau \sim \pi(x, \tau)} [\log Z^\star + \log p^\star(x, \tau)] - H(\pi)$$

$$\propto \exp(\log Z^\star + \log \pi^\star(x, \tau))$$

meaning $\arg\max_{\pi} \max_Z \mathcal{J}_q(\pi, Z) = \arg\min_{\pi} \psi^*(p_{\text{data}} \cdot q - \pi) + K(\pi, q)$.

## A.6 PROPOSITION 4

Starting from Equation 3, we have

$$\arg\max_{\pi, Z} \mathbb{E}_{\substack{x \sim p_{\text{data}}(x) \\ \tau \sim q(\tau|x)}} [\hat{r}_\pi(x; \tau)] - \mathbb{E}_{x, \tau \sim \tilde{\pi}(x, \tau)} [\hat{r}_\pi(x; \tau)] - \psi(\hat{r}_\pi)$$

with the understanding $\tilde{\pi} = \arg\min_{\tilde{\pi}'} \hat{L}_q(\pi, Z, \tilde{\pi}')$. Using the regularizer of the form $\psi(\hat{r}_\pi) = \alpha \mathbb{E}_{d_{\text{mix}}}[(\hat{r}_\pi(x; z) - r_{\text{prior}}(x))^2]$, we have

$$= \arg\max_{\pi, Z} \left( \mathbb{E}_{\substack{x \sim p_{\text{data}}(x) \\ \tau \sim q(\tau|x)}} [\hat{r}_\pi(x; \tau)] - \mathbb{E}_{x, \tau \sim \tilde{\pi}(x, \tau)} [\hat{r}_\pi(x; \tau)] - \alpha \mathbb{E}_{d_{\text{mix}}}[(\hat{r}_\pi(x; \tau) - r_{\text{prior}}(x))^2] \right)$$

$$= \arg\max_{\pi, Z} \left( \mathbb{E}_{\substack{x \sim p_{\text{data}}(x) \\ \tau \sim q(\tau|x)}} \left[ -\frac{\alpha}{2} \hat{r}_\pi(x; \tau)^2 + \hat{r}_\pi(x; \tau) + \alpha \hat{r}_\pi(x; \tau) r_{\text{prior}}(x) \right] \right.$$

$$\left. + \mathbb{E}_{x, \tau \sim \tilde{\pi}(x, \tau)} \left[ -\frac{\alpha}{2} \hat{r}_\pi(x; \tau)^2 - \hat{r}_\pi(x; \tau) + \alpha \hat{r}_\pi(x; \tau) r_{\text{prior}}(x) \right] \right)$$

$$= \arg\max_{\pi, Z} \left( -\frac{\alpha}{2} \mathbb{E}_{\substack{x \sim p_{\text{data}}(x) \\ \tau \sim q(\tau|x)}} \left[ \hat{r}_\pi(x; \tau)^2 - \frac{2}{\alpha} \hat{r}_\pi(x; \tau)(\alpha r_{\text{prior}}(x) + 1) \right] \right.$$

$$\left. - \frac{\alpha}{2} \mathbb{E}_{x, \tau \sim \tilde{\pi}(x, \tau)} \left[ \hat{r}_\pi(x; \tau)^2 - \frac{2}{\alpha} \hat{r}_\pi(x; \tau)(\alpha r_{\text{prior}}(x) - 1) \right] \right)$$

$$= \arg\min_{\pi, Z} \left( \mathbb{E}_{\substack{x \sim p_{\text{data}}(x) \\ \tau \sim q(\tau|x)}} \left[ \left( \hat{r}_\pi(x; \tau) - r_{\text{prior}}(x) - \frac{1}{\alpha} \right)^2 \right] \right.$$

$$\left. + \mathbb{E}_{x, \tau \sim \tilde{\pi}(x, \tau)} \left[ \left( \hat{r}_\pi(x; \tau) - r_{\text{prior}}(x) + \frac{1}{\alpha} \right)^2 \right] \right)$$

$$= \arg\min_{\pi, Z} \mathcal{L}_q^{\text{TB}}(\pi, Z)$$

where $d_{\text{mix}} = \frac{1}{2}(p_{\text{data}} \cdot q + \pi)$, and $\hat{r}_\pi(x;\tau) = \log Z + \log p(x,\pi) - \log q(\pi|x)$. Since $\tilde{\pi} = \arg\min_{\tilde{\pi}'} \hat{L}_q(p, Z, \tilde{\pi}')$, the sampling distribution of the second expectation reduces to $\tilde{\pi} = \pi$ by Lemma 2. This chain of equalities shows that minimizing $\mathcal{L}_q^{\text{TB}}(\pi, Z)$ yields the same optimal pair $(\pi, Z)$ as maximizing $\hat{L}_q(\pi, Z, \tilde{\pi})$. Finally, since $\tilde{\pi}$ is optimal with respect to $\hat{L}_q$, we have $\hat{L}_q(\pi, Z, \tilde{\pi}) = \mathcal{J}_q(\pi, Z)$ by Proposition 2, establishing that $\arg\min_{\pi,Z} \mathcal{L}_q^{\text{TB}}(\pi, Z) = \arg\max_{\pi,Z} \mathcal{J}_q(\pi, Z)$.

### A.7 PROPOSITION 5

Using the variational form of $\chi^2$-divergence (with $f(u) = (u-1)^2$ and $f^*(u) = \frac{1}{4}u^2 + u$; see Appendix B), and $d_{\text{mix}}(x,\tau) = \frac{1}{2}(p_{\text{data}}(x)q(\tau|x) + \pi(x,\tau))$, we have:

$$\min_\pi 2\chi^2(p_{\text{data}} \cdot q \| d_{\text{mix}}) + K(\pi, q)$$

$$= \min_\pi \max_g \mathbb{E}_{\substack{x \sim p_{\text{data}}(x) \\ \tau \sim q(\tau|x)}}[2g(x,\tau)] - \mathbb{E}_{x,\tau \sim d_{\text{mix}}}\left[\frac{1}{2}g(x,\tau)^2 + 2g(x,\tau)\right] + K(\pi, q)$$

$$= \min_\pi \max_g \mathbb{E}_{\substack{x \sim p_{\text{data}}(x) \\ \tau \sim q(\tau|x)}}[g(x,\tau)] - \mathbb{E}_{x,\tau \sim \pi(x,\tau)}[g(x,\tau)] - \mathbb{E}_{x,\tau \sim d_{\text{mix}}}\left[\frac{1}{2}g(x,\tau)^2\right] + K(\pi, q)$$

$$= \min_\pi \max_{\tilde{g}} \mathbb{E}_{\substack{x \sim p_{\text{data}}(x) \\ \tau \sim q(\tau|x)}}[\tilde{g}(x,\tau)] - \mathbb{E}_{x,\tau \sim \pi(x,\tau)}[\tilde{g}(x,\tau)] - \mathbb{E}_{x,\tau \sim d_{\text{mix}}}\left[\frac{1}{2}(\tilde{g}(x,\tau) - c)^2\right] + K(\pi, q)$$

$$= \max_{\tilde{g}} \min_\pi \mathbb{E}_{\substack{x \sim p_{\text{data}}(x) \\ \tau \sim q(\tau|x)}}[\tilde{g}(x,\tau)] - \mathbb{E}_{x,\tau \sim \pi(x,\tau)}[\tilde{g}(x,\tau)] - \psi(\tilde{g}) + K(\pi, q)$$

where $\tilde{g}(x,\tau) = g(x,\tau) + c$ for some constant $c$, and $\psi(\tilde{g}) = \mathbb{E}_{x,\tau \sim d_{\text{mix}}}[\frac{1}{2}(\tilde{g}(x,\tau) - c)^2]$. To change the min-max order, we used saddle point property. The last equation has the same form as Equation 3 under the correspondence $\tilde{g}(x,\tau) = \hat{r}_\pi(x;\tau)$. Since $\hat{r}_\pi(x;\tau)$ contains the $\log Z$ term, its range is unbounded and the constant term can be absorbed, allowing us to substitute $\hat{r}_\pi$ for $\tilde{g}$. By Proposition 2 and 4, this is then equivalent to Equation 5.

### A.8 PROPOSITION 6

We first show that the negative ELBO combined with the DB objective is equivalent to Equation 6. By Proposition 2, the ELBO objective can be written as:

$$\text{ELBO}(\pi, q) = \mathbb{E}_{\substack{x \sim p_{\text{data}}(x) \\ \tau \sim q(\tau|x)}}[\hat{r}_\pi(x;\tau)] - \mathbb{E}_{x,\tau \sim \tilde{\pi}(x,\tau)}[\hat{r}_\pi(x;\tau)],$$

where $\tilde{\pi}(x,\tau) = \arg\min_{\pi'}[\hat{r}_\pi(x;\tau)] + K(\pi', q)$. We decompose the estimated reward $\hat{r}_\pi$ into a sum of per-transition rewards:

$$\hat{r}_\pi(x;\tau) = \log Z + \log \pi(x,\tau) - \log q(\tau|x) = \sum_{t=1}^T \log \frac{F(s_{t-1})\pi(s_t|s_{t-1})}{F(s_t)q(s_{t-1}|s_t)} + \log F(s_T)$$

where $F : \mathcal{S} \to \mathbb{R}$ is the state-flow function, with $F(s_0)$ defined as $\log Z + \log \pi_0(s_0)$ and $F(s_T) = r_{\text{prior}}(s_T)$. Define $\delta(s_{t-1}, s_t) = \log \frac{F(s_{t-1})\pi(s_t|s_{t-1})}{F(s_t)q(s_{t-1}|s_t)}$. Analogous to Equation 3, the ELBO combined with DB can be expressed as $\text{ELBO}(\pi, q) - \psi_{\text{DB}}(\delta)$ where:

$$\text{ELBO}(\pi, q) = \mathbb{E}_{\substack{x \sim p_{\text{data}}(x) \\ \tau \sim q(\tau|x)}}\left[\sum_{t=1}^T \delta(s_{t-1}, s_t) - r_{\text{prior}}(s_T)\right] - \mathbb{E}_{x,\tau \sim \tilde{\pi}(x,\tau)}\left[\sum_{t=1}^T \delta(s_{t-1}, s_t) - r_{\text{prior}}(s_T)\right]$$

and

$$\psi_{\mathrm{DB}}(\delta) = \mathbb{E}_{x,\tau \sim d_{\mathrm{mix}}(x,\tau)} \left[ \alpha \sum_{t=1}^{T} \delta(s_{t-1}, s_t)^2 \right].$$

Since $\tilde{\pi} = \pi$ by Lemma 1, its state-transition distribution is given by $\rho_\pi$. Thus, the objective can be equivalently rewritten in terms of state-transition distribution as follows:

$$\mathbb{E}_{s,s' \sim \rho_q(s,s')} \left[ \delta(s, s') \right] - \mathbb{E}_{s,s' \sim \rho_\pi(s,s')} \left[ \delta(s, s') \right] - \alpha \mathbb{E}_{s,s' \sim \rho_{\mathrm{mix}}} [\delta(s, s')^2], \tag{7}$$

where $\rho_{\mathrm{mix}} = \frac{1}{2}(\rho_q + \rho_\pi)$ and the $r_{\mathrm{prior}}(s_T)$ terms in ELBO are ignored, as it does not affect the optimization. After algebraic manipulation (similar to Appendix A.6), we obtain

$$(7) = -\frac{\alpha}{2} \mathbb{E}_{s,s' \sim \rho_q(s,s')} \left[ \left( \delta(s, s') - \frac{1}{\alpha} \right)^2 \right] - \frac{\alpha}{2} \mathbb{E}_{s,s' \sim \rho_\pi(s,s')} \left[ \left( \delta(s, s') + \frac{1}{\alpha} \right)^2 \right] + \mathrm{constant},$$

which is equivalent to Equation 6 when the maxization problem is reformulated as a minimization problem with $r_\alpha = 1/\alpha$.

Next we proceed to prove that optimizing $\mathcal{L}_q^{\mathrm{DB}}$ is equivalent to minimizing $\chi^2(\rho_q \| \rho_{\mathrm{mix}}) + k(\pi, q)$. Following similar arguments as in Appendix A.7, we use the variational form of $\chi^2$-divergence:

$$\min_\pi 2\chi^2(\rho_q \| \rho_{\mathrm{mix}}) + k(\pi, q)$$

$$= \min_\pi \max_g \mathbb{E}_{s,s' \sim \rho_q(s,s')} [2g(s, s')] - \mathbb{E}_{s,s' \sim \rho_{\mathrm{mix}}(s,s')} \left[ \frac{1}{2} g(s, s')^2 + 2g(s, s') \right] + k(\pi, q)$$

$$= \min_\pi \max_g \mathbb{E}_{s,s' \sim \rho_q(s,s')} [g(s, s')] - \mathbb{E}_{s,s' \sim \rho_\pi(s,s')} [g(s, s')] - \mathbb{E}_{s,s' \sim \rho_{\mathrm{mix}}} \left[ \frac{1}{2} g(s, s')^2 \right] + k(\pi, q)$$

$$= \max_g \min_\pi \mathbb{E}_{s,s' \sim \rho_q(s,s')} [g(s, s')] - \mathbb{E}_{s,s' \sim \rho_\pi(s,s')} [g(s, s')] - \psi(g) + k(\pi, q)$$

$$= \max_g \mathbb{E}_{s,s' \sim \rho_q(s,s')} [g(s, s')] - \mathbb{E}_{s,s' \sim \rho_{\pi_g}(s,s')} [g(s, s')] - \psi(g)$$

where $\psi(g) = \mathbb{E}_{\rho_{\mathrm{mix}}} \left[ \frac{1}{2} g(s, s')^2 \right]$ and $\pi_g(x, \tau) \propto q(\tau|x) \exp(\sum g(s, s'))$. By interpreting $g(s, s')$ as $\delta(s, s')$ recovers $\mathrm{ELBO}(\pi, q) - \psi_{\mathrm{DB}}(\delta)$ under the setting $\alpha = 1/2$ and constant $r_{\mathrm{prior}}$.

# B  STATISTICAL DIVERGENCES

A broad family of divergences can be expressed as $f$-divergences, defined as follows:

$$D_f(p\|q) = \mathbb{E}_{x \sim q(x)} \left[ f\left( \frac{p(x)}{q(x)} \right) \right],$$

where $f$ is a convex, lower-semicontinuous function with $f(1) = 0$. The variational form of $f$-divergences is given as following (Nguyen et al., 2010):

$$D_f(p\|q) = \sup_{c \in \mathcal{C}} \mathbb{E}_{x \sim p(x)}[c(x)] - \mathbb{E}_{x \sim q(x)}[f^*(c(x))]$$

$$= \sup_{c \in \mathcal{C}} \mathbb{E}_{x \sim p(x)}[c(x)] - \mathbb{E}_{x \sim q(x)}[c(x)] - \underbrace{\mathbb{E}_{x \sim q(x)}[f^*(c(x)) - c(x)]}_{\psi_f(c)}$$

$$= \psi_f^*(p - q)$$

where $f^*$ is the convex conjugate of function $f$. Interpreting $p(x) = \pi_{\mathcal{X}}$, $q(x) = p_{\mathrm{data}}(x)$ and $c(x) = -r(x)$, we recover Equation 2 with $K$ removed.

## C   GRADIENT ANALYSIS

In this section, we show that the regularized ELBO is equivalent to TBIL by demonstrating that their gradients coincide. We assume $\pi_\theta$, $q_\phi$, and $Z_\gamma$ are parameterized functions. First, we write the ELBO and TB objectives in terms of these parameterizations:

$$L_{\mathrm{ELBO}}(\theta, \phi) = \mathbb{E}_{x \sim p_{\mathrm{data}}(x), \tau \sim q_\phi(\tau|x)} \left[ \log \frac{\pi_\theta(x, \tau)}{q_\phi(\tau|x)} \right],$$

and

$$\mathrm{TB}(x, \tau; \theta, \phi, \gamma) = \left( \log \frac{Z_\gamma \pi_\theta(x, \tau)}{q_\phi(\tau|x)} - r(x) \right)^2.$$

For convenience, we define

$$\delta_{\mathrm{TB}}(r) = \log \frac{Z_\gamma \pi_\theta(x, \tau)}{q_\phi(\tau|x)} - r(x).$$

Taking gradients with respect to $\theta$, we have

$$\nabla_\theta L_{\mathrm{ELBO}}(\theta, \phi) = \mathbb{E}_{x \sim p_{\mathrm{data}}(x), \tau \sim q_\phi(\tau|x)} \left[ \nabla_\theta \log \pi_\theta(x, \tau) \right],$$

and

$$\nabla_\theta \delta_{\mathrm{TB}}^2(r) = \left( \log \frac{Z_\gamma \pi_\theta(x, \tau)}{q_\phi(\tau|x)} - r(x) \right) \nabla_\theta \log \pi_\theta(x, \tau)$$
$$= \delta_{\mathrm{TB}}(r) \, \nabla_\theta \log \pi_\theta(x, \tau).$$

We will use the standard property that, under the policy distribution, any constant baseline can be subtracted inside the expectation, since

$$\mathbb{E}_{\pi_\theta}[\nabla_\theta \log \pi_\theta(x, \tau)] = 0.$$

Thus,

$$\mathbb{E}_{x, \tau \sim \pi_\theta(x, \tau)}[\nabla_\theta \delta_{\mathrm{TB}}^2(r)] = \mathbb{E}_{x, \tau \sim \pi_\theta(x, \tau)} \left[ \left( \delta_{\mathrm{TB}}(r) - b \right) \nabla_\theta \log \pi_\theta(x, \tau) \right].$$

Let $d_{\mathrm{mix}}$ denote the mixture distribution between $p_{\mathrm{data}} \cdot q$ and $\pi$. Combining the two objectives yields

$$-\nabla_\theta L_{\mathrm{ELBO}}(\theta, \phi) + \mathbb{E}_{x, \tau \sim d_{\mathrm{mix}}(x, \tau)}[\nabla_\theta \alpha \delta_{\mathrm{TB}}^2(r)]$$
$$= \alpha \mathbb{E}_{x \sim p_{\mathrm{data}}(x), \tau \sim q_\phi(\tau|x)} \left[ (\delta_{\mathrm{TB}}(r) - r_\alpha) \nabla_\theta \log \pi_\theta(x, \tau) \right]$$
$$+ \alpha \mathbb{E}_{x, \tau \sim \pi_\theta(x, \tau)} \left[ \delta_{\mathrm{TB}}(r) \nabla_\theta \log \pi_\theta(x, \tau) \right]$$
$$= \alpha \mathbb{E}_{x \sim p_{\mathrm{data}}(x), \tau \sim q_\phi(\tau|x)} \left[ (\delta_{\mathrm{TB}}(r) - r_\alpha) \nabla_\theta \log \pi_\theta(x, \tau) \right]$$
$$+ \alpha \mathbb{E}_{x, \tau \sim \pi_\theta(x, \tau)} \left[ (\delta_{\mathrm{TB}}(r) + r_\alpha) \nabla_\theta \log \pi_\theta(x, \tau) \right]$$
$$= \alpha \mathbb{E}_{x \sim p_{\mathrm{data}}(x), \tau \sim q_\phi(\tau|x)} \left[ \nabla_\theta \delta_{\mathrm{TB}}^2(r + r_\alpha) \right] + \alpha \mathbb{E}_{x, \tau \sim \pi_\theta(x, \tau)} \left[ \nabla_\theta \delta_{\mathrm{TB}}^2(r - r_\alpha) \right],$$

where $r_\alpha = 1/\alpha$. In the second equality, $r_\alpha$ is used as a baseline. The final expression coincides with the gradient of the TBIL objective, up to a constant scaling factor. The gradient with respect to $\phi$ can be derived analogously. Finally, note that the $\log Z_\gamma$ term acts only as a baseline and therefore does not affect the gradient. Consequently, the TBIL and ELBO objectives yield identical gradients, implying that they induce the same policy.

## D  CONNECTIONS TO ENERGY-BASED MODELS

Energy-based models (EBMs) define probability distributions by assigning an unnormalized energy score to each configuration, with lower energies corresponding to higher probabilities. Formally, an EBM specifies a distribution over a space $\mathcal{X}$ as $p(x) = \frac{1}{Z}\exp(-\mathcal{E}(x))$, where $-\mathcal{E}$ is the energy function and $Z = \int \exp(-\mathcal{E}(x))dx$ is the normalizing constant. For consistency with our framework, we set $\mathcal{E}(x) = -r(x)$, so that the distribution can be expressed as $\frac{1}{Z}\exp(r(x))$. EBMs train the function $r$ via maximum likelihood estimation, i.e., by maximizing $\mathbb{E}_{x\sim p_{\text{data}}(x)}[\log\frac{1}{Z}\exp(r(x))]$ with respect to $r$. This objective can be written as:

$$\mathbb{E}_{x\sim p_{\text{data}}(x)}[\log p(x)] = \mathbb{E}_{x\sim p_{\text{data}}(x)}[r(x)] - \log Z$$

$$= \mathbb{E}_{x\sim p_{\text{data}}(x)}[r(x)] - \log\int\exp(r(x))dx$$

$$= \mathbb{E}_{x\sim p_{\text{data}}(x)}[r(x)] - \log\mathbb{E}_{x\sim w(x)}\left[\frac{\exp(r(x))}{w(x)}\right]$$

$$\leq \mathbb{E}_{x\sim p_{\text{data}}(x)}[r(x)] - \mathbb{E}_{x\sim w(x)}\left[\log\frac{\exp(r(x))}{w(x)}\right]$$

$$= \mathbb{E}_{x\sim p_{\text{data}}(x)}[r(x)] - \mathbb{E}_{x\sim w(x)}[r(x)] - H(w)$$

where $w$ is an auxiliary distribution used for importance sampling to estimate $Z$. The inequality follows from Jensen's inequality, with equality holding when $w(x) \propto \exp(r(x))$. In practice, sampling from $p$ is intractable, and EBMs typically rely on Markov chain Monte Carlo (MCMC) methods to generate approximate samples. In addition, to encourage smoothness in $r$ and improve stability during optimization, it is common to introduce a regularization term $\psi(r)$. The optimization problem then takes the form:

$$\max_r\min_w \mathbb{E}_{x\sim p_{\text{data}}(x)}[r(x)] - \mathbb{E}_{x\sim w(x)}[r(x)] - H(w) - \psi(r)$$

where the maximum likelihood estimation problem is reformulated in terms of $r$, and the auxiliary distribution $w$ serves to approximate the normalizing constant $Z$.

The key idea of EB-GFN (Zhang et al., 2022) is to employ GFlowNets as MCMC samplers, thereby reducing approximation errors. In this framework, the auxiliary distribution $w$ is replaced with a GFlowNet sampler $\pi$, and optimization proceeds by interleaving two steps: (1) training $\pi$ to approximate the terminal distribution $\pi_{\mathcal{X}}(x) \propto \exp(r(x))$, and (2) using $\pi$ as a proposal distribution to train $r$. In addition, using an L2 regularization term corresponds to $\psi_{\text{TB}}$ without the prior term $r_{\text{prior}}$, yielding an optimization problem equivalent to Equation 2. However, upon inspecting the source code of EB-GFN, we observed that although L2 regularization is implemented, it does not seem to have been applied. This is equivalent to taking the limit $r_\alpha \to \infty$ in our algorithms. In practice, however, data samples are available only as a finite dataset, and without regularization the GFlowNet is forced to exactly reproduce those datapoints. As a result, the learned policy effectively collapses to sampling directly from the training dataset.

Additional differences from our approach lies both in the optimization procedure and in the focus of the work. In EB-GFN, $r$ and the GFlowNet sampler $\pi$ are trained in alternating steps, with $\pi$ first optimized to approximate $\pi_{\mathcal{X}}(x) \propto \exp(r(x))$, and $r$ subsequently updated using $\pi$ as a proposal distribution. In contrast, we reparameterize $r$ directly in terms of $\pi$, thereby removing the need for this second step and eliminating the alternating optimization. Furthermore, our analysis emphasizes the theoretical connections with MaxEnt IRL, whereas EB-GFN primarily focuses on reducing sampler approximation errors using GFlowNet sampler.

## E  IMPACT OF TERMINAL REWARD

Prior work in imitation learning (Ho & Ermon, 2016; Fu et al., 2017; Garg et al., 2021) often assigned zero rewards to the absorbing state, inadvertently introducing termination or survival bias.

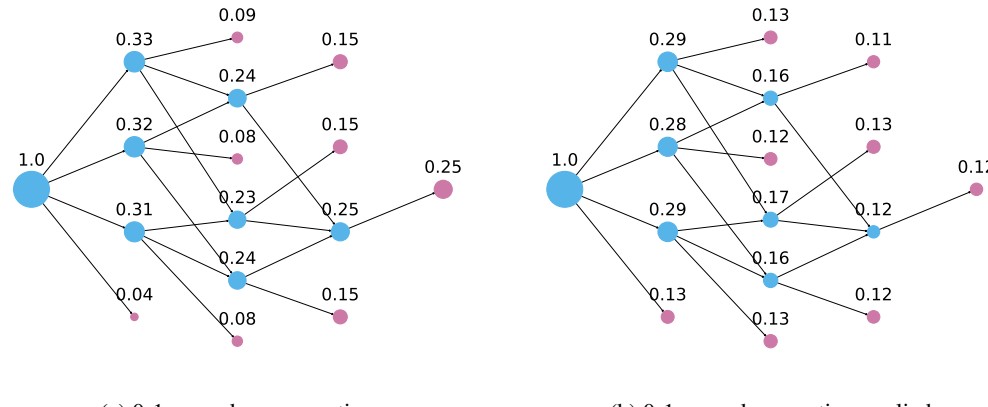

(a) 0-1 reward, no correction

(b) 0-1 reward, correction applied

Figure 4: Illustrative experiments in the bit-flip environment using DBIL. Circle sizes and the numbers indicate state visitation probabilities, while terminal states are highlighted in magenta. The target distribution $p_{\text{data}}(x)$ is uniform, so ideally the visitation probabilities should also be uniform. **(Left)** Zero rewards are assumed for absorbing states. In this case, visitation probabilities correlate with trajectory lengths. **(Right)** With the terminal-state correction applied, visitation probabilities become uniform across terminal states.

This bias arises from improper handling of the absorbing state (Kostrikov et al., 2018; Al-Hafez et al., 2023) and is distinct from the entropy bias discussed in Section 3.1. Our experiments in the Pascal's triangle environment does not have this issue, since the horizon length is fixed.

When the horizon length varies, however, DBIL requires an additional adjustment at the end of trajectories. The reason is that TBIL distinguishes data samples from policy samples using a single reward of $\pm r_\alpha$ applied at the trajectory level, whereas DBIL distributes this adjustment across every transition. As a result, DBIL accumulates a total bonus of $\pm T r_\alpha$ for trajectories of length $T$. This creates a bias when trajectory lengths differ, since longer trajectories automatically accrue larger bonuses (or penalties), even if they terminate in the same outcome. A common workaround is to pad shorter trajectories with dummy absorbing transitions satisfying $\pi(s'|s) = q(s|s') = 1$, so that all trajectories effectively share a fixed horizon. In practice, however, this approach is inefficient, as it introduces unnecessary computations beyond the natural termination point, despite the stopping condition already being known.

The symmetric reward scheme $\pm r_\alpha$ mitigates this issue to some extent, since the bonuses assigned to data samples and policy samples can partially offset each other (and cancel out completely when $p_{\text{data}}(x)q(\tau|x) = \pi(x, \tau)$). Nevertheless, variable horizon lengths still introduce bias, as longer trajectories accumulate larger absolute bonuses. To correct for this, we assign an additional terminal reward that compensates for the missing steps. Concretely, when a trajectory terminates at step $t < T$, we add $\pm (T - t)r_\alpha$ at the terminal state. This adjustment ensures that every trajectory, regardless of its length, accumulates the same total bonus as a trajectory of horizon $T$. In practice, this amounts to padding early-terminating trajectories not with dummy transitions, but with a single corrective reward at termination, thereby avoiding unnecessary computational overhead while maintaining consistency across different horizon lengths.

Figure 4 illustrates the effect of trajectory length in the bit-flip environment, comparing results with and without the proposed correction. In this environment, the initial state is $[0, 0, 0, 0]$, and the policy flips one bit at a time until reaching a terminal state where the last bit is flipped (e.g., $[0, 0, 0, 1]$). Without correction, longer trajectories accumulate larger cumulative $\pm r_\alpha$ bonuses, causing terminal states with longer paths to receive high visitation probabilities. With the terminal-state correction applied, an additional reward of $\pm (T - t)r_\alpha$ is given when a trajectory ends at step $t$, compensating for the difference in horizon length. This adjustment ensures that all terminal states are visited with approximately equal probability, consistent with the uniform target distribution $p_{\text{data}}(x)$.

Table 3: Probability of generating blocks for each method.

|  | 1110 | 0011 | 0110 | 1100 | 0100 |
|---|---|---|---|---|---|
| DBIL ($r_\alpha = 1$) | $0.20 \pm 0.00$ | $0.20 \pm 0.01$ | $0.20 \pm 0.01$ | $0.19 \pm 0.01$ | $0.20 \pm 0.01$ |
| Combined ($r_\alpha = 1$) | $0.21 \pm 0.01$ | $0.19 \pm 0.01$ | $0.20 \pm 0.01$ | $0.21 \pm 0.00$ | $0.19 \pm 0.00$ |
| SQIL ($r_\alpha = 5$) | $0.22 \pm 0.01$ | $0.20 \pm 0.00$ | $0.21 \pm 0.01$ | $0.19 \pm 0.01$ | $0.18 \pm 0.01$ |

# F  ADDITIONAL EXPERIMENTAL DETAILS AND RESULTS

## F.1  GENERATIVE IMITATION LEARNING

**Experimental settings**  The task is to generate a binary sequence by flipping one bit at a time until the `stop` action is selected, with the last bit reserved for this `stop` signal. As in Malkin et al. (2022a), data samples are constructed by randomly concatenating four blocks drawn from the set `0100,1100,0110,0011,1110`, which imposes structure on $p_{\text{data}}$. Consequently, the sequence length is 17 bits in total, yielding $|\mathcal{S}| = 131,072$ states overall and $|\mathcal{X}| = 65,536$ distinct terminal states.

We parameterize the functions $(\pi, F, Q)$ using a two-layer multilayer perceptron (MLP) with 64 hidden units per layer. The key difference between SQIL and DBIL, other than the reward baseline, is the parameterization of functions: DBIL is parameterized by both $F$ and $\pi$, while SQIL relies solely on a soft-$Q$ function (corresponding to $\log F + \log \pi$). In practice, however, we found training a single $Q$ network to be unstable. To address this, we introduced target networks, resulting in an effective parameter size comparable to DBIL. The experiments are run 3 times for each method.

**Results on the entropy bias**  We sampled 5,000 terminal states from each model, yielding 20,000 blocks in total. To evaluate the learned distributions, we measured the frequency of generating the component blocks `0100,1100,0110,0011,1110`. Standard SQIL places high probability on blocks containing more `1`'s, since these lead to terminal states with a larger number of trajectories. In contrast, our method and the combined variant produce block frequencies that are approximately uniform, aligning more closely with the target data distribution.

## F.2  ENERGY-BASED MODELING

**Experimental settings**  We closely follow the experimental setup of Zhang et al. (2022), with the only modification being the treatment of the L2 regularization term. Specifically, for EB-GFN we add L2 regularization with coefficient $\alpha = 0.1$, while for TBIL we set $r_\alpha = 10$. For the combined method, we adopt intermediate values, i.e., $\alpha = 0.2$ and $r_\alpha = 5$. The negative log-likelihood (NLL) is computed following the procedure in Zhang et al. (2022):

$$\mathbb{E}_{\tau \sim q(\tau|x)} \left[ \frac{\pi(x, \tau)}{q(\tau|x)} \right] \approx \frac{1}{M} \sum \frac{\pi(x, \tau)}{q(\tau|x)}$$

where we set $M = 20$. While all methods share the same number of parameters and model architecture for GFlowNets, TBIL does not rely on an explicit energy function, resulting in fewer effective parameters for the task. We evaluated the model every 2,000 steps, and Table 1 reports the best NLL achieved within 100,000 training steps.

**Convergence speed**  We compare the convergence behavior of EB-GFN, TBIL, and their combination in terms of negative log-likelihood (NLL) and the number of model updates. As shown in Figure 5, EB-GFN converges more slowly, possibly because the reward function provides weak training signals in the early stages of optimization. In contrast, TBIL and the combined method converge substantially faster, as the reward is reparameterized directly in terms of the policy and normalization constant. Moreover, EB-GFN requires separate optimization steps, which further increases the time needed for each GFlowNet update. Also see Figure 6 for the visualization of intermediate samples generated by TBIL and EB-GFN.

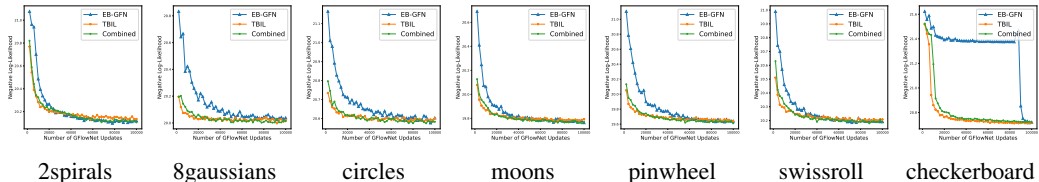

| 2spirals | 8gaussians | circles | moons | pinwheel | swissroll | checkerboard |

Figure 5: Convergence speed measured in terms of negative log-likelihood (NLL) and number of model updates.

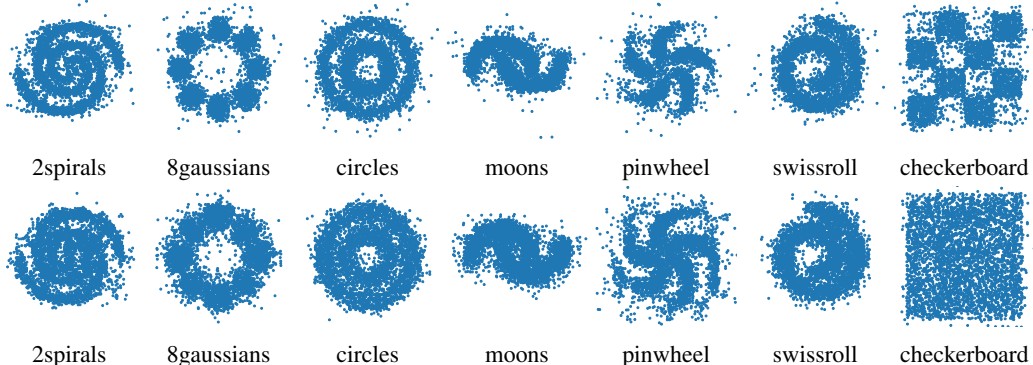

| 2spirals | 8gaussians | circles | moons | pinwheel | swissroll | checkerboard |

| 2spirals | 8gaussians | circles | moons | pinwheel | swissroll | checkerboard |

Figure 6: Visualization of samples generated by GFlowNets after 10k updates. **Top:** TBIL **Bottom:** EB-GFN.

### F.3 OFFLINE LEARNING

**Experimental settings**  We adapt our algorithms to offline RL by setting $r_\alpha$ to a proxy model. For the GFN baseline ($r_\alpha = 0$), we strictly follow the official implementation from Shen et al. (2023) without modification, while our conservative GFN ($r_\alpha = 0.5$) is reimplemented by ourselves. All experiments are conducted under the same training regime for fairness: 25,000 training iterations, 16 training samples (8 on-policy samples + 8 offline data). The offline dataset, used both for training proxy models and constraining the policy distribution, is normalized using standard normalization. For the proxy model, we trained a gradient boosted regressor on the rewards of each task. Hyperparameters were selected using 5-fold cross-validation with grid search, optimizing for mean validation $R^2$. The final model was then retrained on the entire training set using the best hyperparameter configuration.

We evaluated offline learning on three biological sequence design tasks: DNA (TFBind10, TFBind8), and molecules (sEH). These tasks can be formulated as sequence-generation problems under a prepend–append action space. TFBind8 involves generating DNA strings of length 8 over 4 nucleotides ($|\mathcal{X}| = 65,536$), where the reward is the wet-lab measured binding activity to the human transcription factor SIX6 (Trabucco et al., 2022). TFBind10 is the same as TFBind8 but with length 10 ($|\mathcal{X}| = 1,048,576$). The sEH task is to generate molecules that bind to soluble epoxide hydrolase (sEH). Molecules are assembled from 18 building blocks with 2 stems each, using 6 blocks ($|\mathcal{X}| = 34,012,224$). The reward is the predicted binding affinity to the sEH protein from a proxy model trained with AutoDock outputs. The hyperparameters of all three tasks are identical to those used in Shen et al. (2023), except for the number of training rounds and the training sample size.

**Role of offline data**  We evaluated the impact of offline data quality under different dataset settings. For the Table 2 experiments, we constructed two training datasets by randomly sampling 1000 objects from the top 5% and 15% of each reward distribution. Table 2 reports the rewards mean of the top-scoring samples generated by the fully trained models. Interestingly, when the data quality constraint was relaxed (Top 15%), the standard GFN sometimes achieved a higher true reward mean than the conservative GFN on the sEH task. We also observed that policy-generated sample diversity decreased in the more restrictive Top 5% setting. Diversity was quantified using average pairwise distances: Levenshtein distance for DNA sequences and Tanimoto distance between Morgan fingerprints for molecules. To further examine the role of offline data, we compared Top 15% and

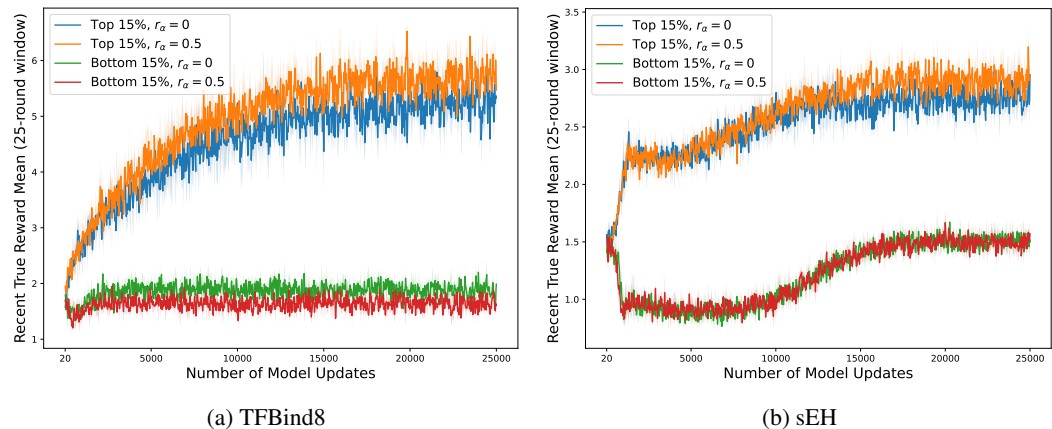

(a) TFBind8                (b) sEH

Figure 7: Impact of offline data quality (Top 15% vs Bottom 15%) on TFBind8 and sEH tasks. We report the true reward mean of generated sequences over training.

Bottom 15% datasets by randomly sampling 500 objects from the top 15% and bottom 15% of the TFBind8 and sEH training datasets. Figure 7 shows the true reward mean of generated sequences during training. 100 on-policy samples were collected every 20 training rounds, and results were averaged over the most recent 25 training rounds. This provides a moving-window view of training quality, capturing short-term fluctuations rather than long-term averages. These results highlight the importance of offline dataset quality: while the conservative GFN consistently performs better with Top 15% data, it underperforms the standard GFN when trained on Bottom 15% data.

## G  THE USE OF LARGE LANGUAGE MODELS

During the preparation of this paper, we made use of a large language model (ChatGPT, OpenAI GPT-5) as a writing and editing assistant. Its role was limited to:

- Proofreading and polishing text: improving grammar, readability, and stylistic consistency.
- Paraphrasing and rephrasing: providing alternative wordings for sentences and figure captions while maintaining technical accuracy.
- Consistency checks: ensuring consistent terminology, notation, and tone across sections.

All mathematical derivations, algorithmic formulations, experimental design, and scientific claims were developed and validated by the authors. The LLM did not generate new research ideas or contribute original technical content.

