# OpenReview forum: "Inverse GFlowNets for Generative Imitation Learning"
_ICLR.cc/2026/Conference — ICLR 2026 Conference Withdrawn Submission_

### Official Review · Reviewer_2ACN · 2025-10-21

**Soundness:** 2
**Presentation:** 3
**Contribution:** 3
**Rating:** 4
**Confidence:** 3

**Summary:**

The focus of the paper is studying ELBO-based training of sequential generative models through imitation learning paradigm. Building upon the theoretical framework of GFlowNets and their connections to MaxEnt RL, the paper establishes further connections with MaxEnt inverse RL (IRL) and energy-based models (EBMs). Based on the presented connections, the authors derive a form of regularization for ELBO inspired by IRL, also showing that the presented regularized objective is equivalent to a combination of GFlowNet objectives. The approach is experimentally evaluated on 2D synthetic problems, as well as two biological sequence generation problems.

**Strengths:**

I believe that the theoretical parts of this paper present a number of valuable and highly original insights, nicely connecting a very diverse number of topics (generative models, imitation learning, inverse reinforcement learning, GFlowNets, energy-based models) into a coherent methodological picture. In my opinion, deriving regularization techniques for generative model training from reinforcement learning and GFlowNets is an interesting and important research direction. I also found the theoretical and methodological parts of this paper to be very well-written and easy to follow (with some minor exceptions, see Weaknesses).

**Weaknesses:**

In my opinion, the main weakness of this paper is its limited experimental evaluation. It is difficult to derive any conclusions about the practical utility of the proposed method from the presented experimental results. Sections 5.1 and 5.2 present toy synthetic experiments. Even on the toy 2D tasks, all differences in metric in comparison to EB-GFN baseline are very marginal (Table 1). Section 5.3 presents evaluation on two biological sequence generation tasks. In Table 2, the scale of the improvement in these tasks is marginal and comparable to std in many cases, while there are also cases where the approach falls behind the GFlowNet baseline.

In addition, the setup of the biological sequence generation experiment deviates from the setup of generative modeling that the paper claims to be studying. TFBind10 and sEH are tasks of sampling proportionally to the reward studied in GFlowNet literature, and the authors themselves state that the dataset is used to constrain the policy distribution, while the main training signal still comes from the reward. This is confusing and somewhat misleading in my opinion, so either the abstract and intro should be adjusted to accurately reflect the problems and setup the authors are studying in the paper, or the experimental setup for sequence generation tasks should be adjusted itself.

As a suggestion for improvement on this part, applying the proposed regularization technique to diffusion model training would be a great addition to the experimental evaluation in this paper. From what I understand, the methodology in this paper is designed to be directly applicable to any type of diffusion model that uses ELBO as a training objective, and diffusion models themselves can be directly put into the GFlowNet framework [1, 2]. So I suggest to add experiments with either continuous diffusion models in such domains as images or molecules, or discrete diffusion models for text generation.

Next, Section 3.1 is a bit hard to understand in its current form as no section in the Background explains how IRL can be applied to generative modeling. This becomes more clear after reading further sections, so this is a minor issue of mine.

Furthermore, there is a highly relevant paper that studies connections between GFlowNets and generative models and ELBO-based training [1]. It also proposes using a GFlowNet based objective along ELBO for generative model training (Algorithm 1). I suggest that the paper should be referenced and discussed in the text.

There are also works that apply IRL methodology to train energy-based diffusion models [3], which I also suggest referencing and discussing.

Finally, a minor comment of mine. The authors mention that unlike RL, GFlowNets allow the backward policy to be trained jointly with the forward policy (line 111). There is a paper [4] that studies learnable backward policies in GFlowNets from the RL perspective, framing the training as a non-stationary RL problem. This might also be of interest to the authors as they consider trainable backward policies and connections to IRL.


References: \
[1] Zhang et al. Unifying Generative Models with GFlowNets and Beyond. ICML 2022 Workshop \
[2] Lahlou et al. A theory of continuous generative flow networks. ICML 2023 \
[3] Yoon et al. Maximum Entropy Inverse Reinforcement Learning of Diffusion Models with Energy-Based Models. NeurIPS 2024\
[4] Gritsaev et al. Optimizing Backward Policies in GFlowNets via Trajectory Likelihood Maximization. ICML 2025

**Questions:**

1) What is $r_{\operatorname{prior}}$ in each of the presented experiments? I believe that this should be emphasized in the text as this is one of the central parts of the presented algorithm.

2) In Section 3.1 it is stated: "The task is to generate a 17-bit binary sequence by flipping one bit at a time until the stop action is chosen." (line 400). Wouldn't this result in an environment with cycles? Meanwhile, the paper claims to be working in an acyclic setting (line 118).

3) Can ELBO-based training with the addition of TBC loss from [1] also be viewed as a form of Regularized ELBO from Proposition 2?

References:\
[1] Zhang et al. Unifying Generative Models with GFlowNets and Beyond. ICML 2022 Workshop

---

### Official Review · Reviewer_DkuS · 2025-10-27

**Soundness:** 2
**Presentation:** 1
**Contribution:** 2
**Rating:** 2
**Confidence:** 4

**Summary:**

The paper studies sequential generative modeling through imitation learning, arguing that MaxEnt IRL induces an entropy bias unsuited to ELBO‑trained models. It proposes a GFlowNets‑based framework that interprets ELBO as a regularized objective and yields two practical algorithms—Trajectory‑Balance and Detailed‑Balance imitation learning (TBIL/DBIL). Empirically, on synthetic distributions and biological sequence design, the approach mitigates the entropy bias and achieves competitive accuracy and convergence compared to SQIL and EB‑GFN baselines.

**Strengths:**

To my knowledge, the paper is the first one to design the generative imitation learning framework built on GFlowNets. The authors also provide a theoretical analysis of their approach, establishing connections with variational inference and MaxEntropy reinforcement learning.

**Weaknesses:**

1. The paper is not very well written. The exposition is not consequitive enough, especially at the beginning of the main part (sections 2.2 till 3.2). In particular:
- Notation for $\bar{r}$ is overloaded (both per-step and cumulative reward);
- $\psi$ is not properly defined, as many other things related with policies and MDP framework. Only from the context one can infer, that $\psi$ acts $S \times S \to \mathbb{R}$, and $\bar{r}$ is identified with a matrix $S \times S$.
- $p_{\text{data}}$ appears first in section 2.3 and confuses the reader, since there was no discussion about the particular dataset and on the problem setting itself earlier
- Details of section $3.1$ are unclear, and this section comes, perhaps, too early. The authors do not provide sufficient details neither about the Pascal environment, not about the setting used in the experiments and baselines,

2. As the authors acknowledge, the experimental section is rather limited with a lack of larger scale experiments. It is not clear, what is the practical motivation of the method - while the authors claim that suggested approach enables, for example, the joint training of forward and backward processes in GFlowNets, there are no real ablations and comparisons, indicating the advantage of their approach over the baselines on the competitive problems.

3. The are notable issues with empirical evaluation, in addition to the choice of environments itself. Examples from section 5.1 compare only detailed-balance based method (DBIL), not TB. Contrary, in 5.2 they provide results for TBIL only.  It is unclear, which loss (with which empirical tricks: replay buffer, temperature annealing, target networks, ...) were used in Section 5.3. This section is very scarce and does not provide enough information to understand the details of experiment.

4. The metrics for EB-GFN example seems to be worse, compared to the original paper [Zhang et al, 2022]. Why is it so?

**Questions:**

Are there any hypotheses for the methods metrics from Table 2? What should be the interpretation of these numbers, and why particular values of $r_\alpha$ perform differently on different datasets?

---

### Official Review · Reviewer_D8wg · 2025-10-31

**Soundness:** 2
**Presentation:** 3
**Contribution:** 2
**Rating:** 2
**Confidence:** 3

**Summary:**

The authors discuss the limitations of maximum entropy inverse reinforcement learning for sequential generative modelling. To address this, they present an imitation learning approach to model data distributions using the GFlowNets framework. The authors derive an optimization problem analogous to inverse reinforcement learning and interpret it as regularized ELBO objective training, which can be optimized through two competing GFlowNet objectives.

**Strengths:**

1. The paper designs the first generative imitation learning framework built on GFlowNets.
2. The theoretical analysis of the provided approach. Specifically, the derivation of the optimization problem, connection to GFlowNets objectives, and divergence interpretation. It helps to follow the work and emphasizes the contribution.

**Weaknesses:**

1. The experimental section is limited, and the authors agree with it in conclusion. Clearly, there is a lack of large scale experiments, and there is no ablation study on the hyperparameters used.

2. According to the empirical validation, I doubt that the proposed method indeed achieves improved performance and can be practically useful.
In section 5.1 authors benchmark only DBIL, no results for TBIL.
In section 5.2 “We compare TBIL with EB-GFN”, but:
(a) for EB-GFN authors in this paper use L2 regularization, which negatively affects its performance – the default EB-GFN from the original paper beats all methods in every task illustrated in Table 1 (the results in section F.2 are confusing as well);
(b) unclear why the authors do not provide results for DBIL here.
In section 5.3 authors do not even specify what type of loss in GFlowNets used. This section is poorly written, and the information in Appendix F.3 is not enough for a clear understanding. Moreover, in Appendix F.3, the authors mention that they evaluate offline learning on three biological sequence design tasks (1116-1117), but there are results only for two of them (TFBind8 missed). For experiments, the authors use data “from the top 5% and 15%” (465-466) – (a) unclear what the criterion of the top is, but guess it is a reward; (b) unclear why exactly these two numbers were chosen – 5% and 15%.
And on top of that, the benchmarking seems to be poor as other imitation learning methods (only SQIL 2019 in section 5.1) were not taken into account.

**Questions:**

1. How sensitive is the model's performance to the value of r_a?
2. Can you explain the results obtained in Table 2 please, in particular, why one method or another performs better in different settings?
3. Same as weaknesses 2.

**Details Of Ethics Concerns:**

-

---

### Note · Authors · 2025-12-02

**Comment:**

We sincerely appreciate the reviewers’ valuable feedback, especially their recognition of the theoretical contribution of our work and its introduction of the first generative imitation learning framework based on GFlowNets. At the same time, we have realized that the experimental section may be confusing in its current form, and that the paper would benefit substantially from clearer presentation and more comprehensive experiments—improvements that cannot be completed within the rebuttal period. We therefore respectfully withdraw our submission from the review process. After careful consideration, we have decided to revise the work in the future. We appreciate the reviewers’ time and thoughtful evaluation.

Before concluding, we would like to briefly address some of the questions raised by the reviewers, as these clarifications may be of interest.

**D8wg** - We appreciate your thorough evaluation of our empirical section, particularly on missing ablations and unclear experimental details. We compared TBIL with EB-GFN because EB-GFN is originally evaluated only with the TB objective. The experiments in Section 5.3 also use the TBIL objective. We are grateful for your careful scrutiny and will incorporate these insights into a substantial revision.

**DkuS** - We sincerely appreciate your detailed assessment and your helpful comments on unclear aspects of our work. Regarding the EB-GFN results, the weaker performance in our experiments may stem from the addition of L2 regularization, as well as potential differences in unstated hyperparameters despite our best efforts to match the original settings. We will work to make these points more accessible and transparent in future revisions.

**2ACN** - We sincerely appreciate your thoughtful suggestions. Some of the referenced works were not known to us, and they will be extremely helpful for improving the paper. Regarding your question on the 17-bit sequence generation environment, it is acyclic because only 0-bits can be flipped to 1-bits. Concerning the TBC loss, we do not believe it can be viewed as a form of regularized ELBO, as it enforces only a consistency constraint without incorporating any prior, consistent with the statement in the original paper: “The proposed consistency loss objective only assures the balance between the forward and backward trajectories of the GFlowNet model, but receives no signal about information of the target distribution that the GFlowNet desires to match.” We will carefully incorporate your suggestions and expand the experimental scope in a future version of the work.

**Withdrawal Confirmation:**

I have read and agree with the venue's withdrawal policy on behalf of myself and my co-authors.